# Antibiotic Resistance in Bacteria—A Review

**DOI:** 10.3390/antibiotics11081079

**Published:** 2022-08-09

**Authors:** Renata Urban-Chmiel, Agnieszka Marek, Dagmara Stępień-Pyśniak, Kinga Wieczorek, Marta Dec, Anna Nowaczek, Jacek Osek

**Affiliations:** 1Department of Veterinary Prevention and Avian Diseases, Faculty of Veterinary Medicine, University of Life Sciences in Lublin, 20-033 Lublin, Poland; 2Department of Hygiene of Food of Animal Origin, National Veterinary Research Institute, Partyzantów 57, 24-100 Puławy, Poland

**Keywords:** antibiotic resistance, bacteria, resistance genes, antimicrobials

## Abstract

Background: A global problem of multi-drug resistance (MDR) among bacteria is the cause of hundreds of thousands of deaths every year. In response to the significant increase of MDR bacteria, legislative measures have widely been taken to limit or eliminate the use of antibiotics, including in the form of feed additives for livestock, but also in metaphylaxis and its treatment, which was the subject of EU Regulation in 2019/6. Numerous studies have documented that bacteria use both phenotypis and gentic strategies enabling a natural defence against antibiotics and the induction of mechanisms in increasing resistance to the used antibacterial chemicals. The mechanisms presented in this review developed by the bacteria have a significant impact on reducing the ability to combat bacterial infections in humans and animals. Moreover, the high prevalence of multi-resistant strains in the environment and the ease of transmission of drug-resistance genes between the different bacterial species including commensal flora and pathogenic like foodborne pathogens (*E. coli*, *Campylobacter* spp., *Enterococcus* spp., *Salmonella* spp., *Listeria* spp., *Staphylococcus* spp.) favor the rapid spread of multi-resistance among bacteria in humans and animals. Given the global threat posed by the widespread phenomenon of multi-drug resistance among bacteria which are dangerous for humans and animals, the subject of this study is the presentation of the mechanisms of resistance in most frequent bacteria called as “foodborne pathoges” isolated from human and animals. In order to present the significance of the global problem related to multi-drug resistance among selected pathogens, especially those danger to humans, the publication also presents statistical data on the percentage range of occurrence of drug resistance among selected bacteria in various regions of the world. In addition to the phenotypic characteristics of pathogen resistance, this review also presents detailed information on the detection of drug resistance genes for specific groups of antibiotics. It should be emphasized that the manuscript also presents the results of own research i.e., *Campylobacter* spp., *E. coli* or *Enetrococcus* spp. This subject and the presentation of data on the risks of drug resistance among bacteria will contribute to initiating research in implementing the prevention of drug resistance and the development of alternatives for antimicrobials methods of controlling bacteria.

## 1. Introduction

Widespread resistance to antibiotics among bacteria is the cause of hundreds of thousands of deaths every year. The most serious problem is the constantly growing number of bacteria resistant to commonly used antibiotics, including drugs of last resort (vancomycin). The speed with which resistance genes can spread around the world confirms the worrying rise in a problem that affects public health on a global scale and requires international cooperation (Appendix A) In response to the significant increase in the population of multi-drug resistant strains observed worldwide, in 2014 the World Health Organization (WHO) recognized this phenomenon as a major global health threat [1].

Legislative measures have widely been taken to limit or eliminate the use of antibiotics, including in the form of feed additives for livestock, but also as antibacterial agents in metaphylaxis and treatment, which was the subject of EU Regulation 2019/6 [2].

As part of the strategies undertaken to reduce of drug resistance among microorganisms, it is also necessary to increase the research potential in such areas as genetic improvement of animals in order to identify markers associated with increased innate resistance to pathogens, search for new antimicrobial agents, and determine the role of bacteria in the transmission of antibiotic resistance to human and animal microbial flora. The currently implemented strategies for overcoming antibiotic resistance rely on the alternative use of bacteriophages or their enzymes, the development of next-generation vaccines. Also important is the use of new feeding-based regimes for animals with prebiotics, probiotics, bacterial subproducts, and phytobiotics [3]. There is also great interest in proteins and peptides with bactericidal activity synthesized by bacteria, plants, invertebrates, vertebrates and mammals. This solution is based on the use of antimicrobial peptides produced by generally recognized as safe (GRAS) bacteria like *Lactobacillus* spp., *Streptomyces*, *Micrococcus* or yeast *Saccharomyces* and *Candidia* [4].

The fight against the development of antibiotic resistance has conventionally taken place mainly in clinical conditions, and more recently in agriculture as well, with the aim of limiting transmission of resistant bacteria and preventing their selection during treatment with antibiotics. Recent years have seen an increasing understanding of the role of the environment as an important source and pathway of the dissemination of drug resistance among bacteria.

## 2. Mechanisms of Acquisition of Drug Resistance among Bacteria

The simplest type of resistance is a natural lack of susceptibility, called innate resistance. This is a constant trait of a species, strain, or whole group of bacteria. A given microorganism is insensitive to an antibiotic due to its ‘innate’ resistance to certain groups of antibiotics. It may be linked to the absence of a receptor for the antibiotic, low affinity, cell wall impermeability, or enzyme production [5].

Changes in the susceptibility of bacteria can be primary or secondary. Primary resistance arises as a result of a spontaneous mutation and can appear without contact with a drug. This type of resistance is encoded chromosomally and is not transmitted to other bacterial species. The frequency of occurrence of mutated bacteria is low, but in the presence of an antibiotic, mutants have an advantage over the rest of the population, and thus they survive and outnumber susceptible populations. They can spread to other ecological niches in the same individual or can be transferred to other macroorganisms. While defending themselves against antibacterial agents, including antibiotics, in the course of their evolution, bacteria have developed a variety of mechanisms counteracting the effects of antibacterial agents. As a result of the acquisition of resistance genes, bacteria become partly or entirely resistant to a given antibiotic by developing various effector mechanisms [6].

Based on numerous scientific studies conducted from the mid-20th century, a number of mechanisms explaining bacterial resistance to antibiotics have been proposed. Bacteria are currently believed to acquire antibiotic resistance via active removal of the antibiotic from the cell, enzymatic modifications of the antibiotic, modifications of cell components which are the target of the antibiotic, overexpression of an enzyme inactivated by the antibiotic, a change in the permeability of bacteria cell membranes, production of an alternative metabolic pathway, an increase in the concentration of a metabolite which is an antagonist of the antibiotic, a reduction in the amount or activity of an enzyme activating the precursor of the antibiotic, modifications in regulatory systems not associated with the direct mechanism of action of the antibiotic, or a reduction in the demand for the product of the inhibited metabolic pathway [7,8].

Numerous studies have documented that bacteria use two main genetic strategies enabling natural defence against antibiotics: gene mutation, often associated with the mechanism of action of an antibacterial compound, and acquisition of foreign DNA encoding determinants of resistance via horizontal gene transfer [9].

Horizontal gene transfer plays an important role in the spread of both known and new, as yet unidentified resistance genes. This mechanism allows resistance to expand beyond specific clones. In this way gene transfer makes resistance genes available for a much larger number of bacteria, even breaking the species barrier between environmental (non-pathogenic) bacteria and pathogens in a given living environment of microorganisms [10]. The process of horizontal transfer of drug resistance genes between bacteria can take place in any environment where they are present. However, for resistance genes to be transferred horizontally from environmental to pathogenic bacteria, they must at least temporarily be present in the same environment. In addition, horizontal gene transfer is much more likely between bacteria that are closely phylogenetically related [11]. Finally, transfer of genetic material between bacterial cells is induced by stressors such as antibiotics [12], and potentially also metals and biocides [13]. Selection of antibiotics also contributes to the establishment of transferred resistance genes in a new host. Therefore, transfer of resistance to pathogens can be expected to be relatively common between bacteria associated with humans [14], especially during treatment with antibiotics. In contrast, transfer of resistance genes to pathogens from environmental bacteria that occupy a different habitat and are often less closely phylogenetically related would most likely be less common, although environmental stressors can induce horizontal gene transfer to and from (opportunistic) human pathogens in environmental conditions. This means that when a resistance factor is transferred to a human pathogen, there is a greater chance of its further spread between commensals and pathogens than of transfer to another pathogen from environmental bacteria.

Mechanisms leading to secondary resistance, which develop in conditions of contact between the microorganism and an antibacterial drug, are much more complex. The secondary resistance mechanism is extrachromosomal. The genes responsible for this phenomenon are located in small circular molecules of DNA called plasmids in the cytoplasm. One plasmid may contain genes of resistance to several different antimicrobials. Plasmids can transfer genes encoding resistance from one bacterial cell to another. Plasmids are transferred mainly via conjugation and transduction. During conjugation, plasmids are transferred by direct contact between two or more bacterial cells via strands of protein produced by them. Bacteria of different species and genera, often phylogenetically remote, can take part in the conjugation process. Transfer of resistance from saprophytic to pathogenic bacteria in this manner is particularly unfavourable. Transduction is the process of transfer of plasmids from the donor cell to the recipient cell, mediated by bacteriophages (bacterial viruses). After the bacteriophage attaches to a receptor on the cell surface, the DNA is introduced into the bacterium. The bacteriophage exploits the metabolic processes of the cell to replicate viral DNA and produce viral proteins. Following the formation of new bacteriophages inside the bacterial cell, it undergoes lysis—the lytic cycle. Phage DNA can also become incorporated into the bacterial chromosome (prophage), which is called lysogeny [8,15]. Among transposition elements capable of changing places in the genome, we can distinguish insertion sequences (IS) and transposons (Tn). Insertion sequences are DNA segments containing a gene coding for transposase, surrounded on both sides by inverted repeat sequences. This enzyme allows insertion elements to move to any site in DNA. Resistance genes can also be located on transposons, sometimes called ‘jumping genes’. Among transposons (Tn) we can distinguish composite transposons, which consist of two insertion sequences located on either side of genes encoding resistance to antibiotics or other genes not associated with movement of the transposon (e.g., Tn10). In non-composite transposons (type Tn3), genes encoding additional traits are surrounded by short inverted sequences, and transposition is replicative and requires the products of both genes. Conjugative transposons differ from classic transposons in that they can be transferred not only within the DNA of a single cell, but also between cells. They occur in a form integrated with a plasmid or bacterial chromosome. In response to certain signals, these transposons form circular forms that are incapable of replication. The transfer is similar as in the case of conjugative plasmids [8]. In the evolution of multi-drug resistance in bacteria, an important role is also ascribed to integrons, which can be located in both bacterial chromosomes and plasmids. This is a specific self-translocational type of specialized carriers of genetic information, whose special property is the ability to combine resistance genes into cassettes, which are transferred together in this form to recipient cells [7]. 

Given the global threat posed by the widespread phenomenon of multi-drug resistance among bacteria which are dangerous for humans and animals, the subject of this study is the mechanisms of resistance of bacteria isolated from human and animals. This review also presents the percentage range of the occurrence of drug resistance among selected bacteria. 

## 3. Mechanisms of Transfer of Resistance Based on Examples of Various Species of Bacteria 

### 3.1. Campylobacter spp.

*Campylobacter*, especially *C. jejuni*, is recognized as one of the most common causes of food-borne gastroenteritis in humans [16,17]. Poultry has been shown as the most important source of these bacteria and the main transmission route of *Campylobacter* to humans is the handling of contaminated food, especially of chicken meat [16,17,18]. However, ruminants, especially cattle, and pigs are also responsible for human *C. jejuni* and, to a lesser extent *C. coli*, infections [19,20]. The majority of campylobacteriosis cases are usually self-limiting and do not require any antibacterial treatment. However, the medical interventions are needed in very young or elderly patients, pregnant women, or when the complications such as e.g., Guillain-Barré syndrome develop [21]. In these cases, the macrolides (e.g., erythromycin) are usually the antimicrobials of the first choice, whereas fluoroquinolones (e.g., ciprofloxacin) or tetracyclines are alternative options [22].

#### 3.1.1. Resistance to Fluoroquinolones

AMR of *Campylobacter*, especially to fluoroquinolones (e.g., ciprofloxacin), has recently emerged [23]. Moreover, co-resistance to fluoroquinolones and macrolides (e.g., erythromycin), other antimicrobials of choice for treatment of *Campylobacter* infections, has recently been noted [24]. Furthermore, some studies have shown that infections with quinolone- or erythromycin-resistant *Campylobacter* cause longer and more severe disease, as well as an increased risk of an invasive form of the illness or even death [25]. Several molecular mechanisms of antimicrobial resistance have been identified in *Campylobacter* [26,27,28]. One involves mutations in the genetic material which lead to the development of total or at least partial resistance to various antimicrobials [29]. *Campylobacter* can acquire antibiotic resistance-encoding sequences via horizontal transfer of genes from other bacteria of the same or different species [26]. Antimicrobial resistance can also be acquired through spontaneous mutations in genes resulting in genetic markers responsible for chromosomally encoded resistance to fluoroquinolones or macrolides [30]. Mutations responsible for *Campylobacter* resistance to fluoroquinolones are mainly identified in the quinolone resistance-determining region (QRDR) of DNA gyrase (topoisomerase II) encoded by the *gyrA* and *gyrB* genes, responsible for synthesis of two subunits of the enzyme (subunits A and B, respectively) [31]. Point mutations in the *gyrA* sequence at positions Thr-86, Asp-90, and Ala-70 have been linked to fluoroquinolone resistance in *C. jejuni* [32]. Among these, the Thr-86-Ile change in the *gyrA* gene is the most commonly observed mutation responsible for high-level resistance in fluoroquinolone-resistant *Campylobacter*, whereas the Thr-86-Lys and Asp-90-Asn mutations are less common and are associated with intermediate fluoroquinolone resistance [33]. Double point mutations of the *gyrA* gene together with Asp-85-Tyr, Asp-90-Asn, or Pro-104-Ser have also been reported [34]. A high level of resistance to fluoroquinolones in *Campylobacter* can also be due to the synergistic effect of the *gyrA* mutation combined with the action of the CmeABC multidrug efflux pump encoded by an operon consisting of three genes, *cmeA*, *cmeB*, and *cmeC*, responsible for the expression of a periplasmic fusion protein, an inner membrane drug transporter, and an outer membrane protein, respectively [35]. The CmeABC efflux pump is regulated by the CmeR repressor, which is highly conserved in nature. An insertional *cmeR Campylobacter* mutant strain showed overexpression of CmeABC pump components, and consequently a decrease in the intracellular concentration of antibiotic. Furthermore, when this efflux pump had been blocked, the minimum inhibitory concentration (MIC) values for fluoroquinolones (e.g., ciprofloxacin) were reduced to the level identified in susceptible strains, even in the presence of mutations in the *gyrA* gene [32]. Recently, Yao et al. using whole genome sequencing showed that the *cmeABC* genes are globally distributed among human and poultry *C. jejuni* which may be horizontally transferred between strains of different origins [36].

#### 3.1.2. Resistance to Macrolides

The frequency of mutations of the genes responsible for macrolide resistance in *Campylobacter* is much less common than in the case of fluoroquinolone resistance. Moreover, emergence of high-level resistance may require multiple mutation steps; thus, macrolide-resistant *Campylobacter* mutants usually develop more slowly under selective antibiotic pressure than under the influence of fluoroquinolones, so they need prolonged exposure to macrolide antimicrobial agents [37]. Resistance of *Campylobacter* to this class of antibiotics is usually the result of modification of the ribosome target binding site by mutation of 23S rRNA at positions 2074 (A2074C, A2074G, or A2074T), 2075 (A2075G or A2075C), or both of the adenine residues in all three copies of this gene (rrnB operon) [37]. However, high-level macrolide resistance is mainly associated with a modification at A2075G in domain V of 23S rRNA. Such resistance mechanisms to erythromycin have also been shown to correspond with cross-resistance to other macrolides and related drugs of the lincosamide and streptogramin classes [29].

Resistance of *Campylobacter* isolates to macrolides may also be the result of modifications of the ribosomal subunit proteins L4 and L22 of 50S ribosome, encoded by the *rplD* and *rplV* genes, respectively [33]. The G-to-A transition at nucleotide 221 of the *rplD* gene causes a glycine to asparagine substitution at position 74 of the L4 protein sequence [38]. In the case of the L22 protein, the main cause of macrolide resistance is duplication at positions 292 and 256 in the *rplV* gene [39]. Additionally, the presence of the emerging *ermB* gene has been linked to macrolide resistance in *Campylobacter* [30]. This gene encodes an rRNA methylase which produces cross-resistance to macrolides, lincosamides and streptogramin B [40]. The *ermB*-encoded enzyme acts on the 23S rRNA gene by methylating an adenine residue that impedes antibiotic binding to the ribosome [41]. Comparative genomic analysis identified identical *ermB* sequences among *Campylobacter* and other bacterial species from animals and humans, which suggests possible horizontal transfer and wide dissemination of antimicrobial resistance to macrolides, which may reduce the effectiveness of antimicrobial therapy.

#### 3.1.3. Resistance to Tetracyclines

Resistance of *Campylobacter* to tetracyclines is mainly conferred by the *tet(O)* gene located on the pTet plasmid and coding for the ribosomal protection protein *tet(O)*. However, the high-level resistance of these bacteria to tetracycline also depends on the CmeABC efflux pumps, alone or in connection with the plasmid-encoded *tet(O)* gene [42]. The *tet(O)* gene is located on a self-transmissible plasmid of molecular size from 45 to 58 kb [43]. However, location of this sequence on the chromosome has also been reported among tetracycline-resistant *C. jejuni* without the *tet(O)*-positive plasmid [44]. Furthermore, the presence of an insertion element IS607 on the *tet(O)*-carrying plasmid in *Campylobacter*, which was similar to IS607 present on the chromosome of *Helicobacter pylori*, may suggest that mobile genetic elements other than transmissible plasmids may be involved in the acquisition and dissemination of *tet(O)* tetracycline resistance genes in other bacterial species. Sequence analysis of G-C content also suggests that this gene was probably acquired by *Campylobacter* via horizontal gene transfer from *Streptomyces*, *Streptococcus*, or *Enterococcus* spp. Since the conjugative *tet(O)*-carrying plasmid is widely prevalent among *Campylobacter*, it is possible that conjugation has contributed to the spread of tetracycline resistance genes among these and other bacterial species [27].

#### 3.1.4. Resistance to β-Lactams

The above-mentioned multidrug efflux pumps, such as CmeABC, also play a role in the prevalence of resistance of *Campylobacter* to β-lactam antibiotics, e.g., ampicillin [35]. A significant increase in susceptibility to ampicillin has been demonstrated in CmeABC-inactivated *C. jejuni* mutants, and a decrease in susceptibility in CmeABC-overexpressing mutants [45]. Another mechanism of β-lactam resistance in *Campylobacter* is the production of chromosomally encoded β-lactamase OXA-61 [28]. The expression level of this gene modulates the susceptibility of the bacteria to this class of antimicrobials, e.g., a single nucleotide mutation (G–T transversion) in the promoter region of *bla*_OXA-61_ led to overexpression of the gene and consequently to a high increase in β-lactam resistance in *C. jejuni* [46]. *Campylobacter* bacteria are capable of intrinsic production of β-lactamases in the absence of selective (antibiotic) pressure. The *bla*_OXA-61_ gene has been shown to be widely distributed among *Campylobacter* isolated from poultry but has also been identified in isolates from non-food producing animals and environments [47,48]. On the other hand, β-lactam antibiotics are known to have limited efficacy against *Campylobacter*, and resistance to this class of antimicrobials appears to be mediated by both intrinsic resistance and β-lactamase production [26]. Oral β-lactams, such as co-amoxiclav, could be an appropriate choice when *Campylobacter* is resistant to both fluoroquinolones and macrolides [28].

The spread of various genetic determinants responsible for *Campylobacter* resistance to quinolones, macrolides, and tetracyclines is very common and increases the prevalence of antimicrobial-resistant *Campylobacter* isolated from humans, foods of animal origin, and animals (Table 1). Resistance to fluoroquinolones and tetracyclines is very common among clinically- and broiler-associated *Campylobacter*, while prevalence of erythromycin-resistant strains is fairly low, or moderate, in Europe [24]. Resistance to quinolones is an especially important concern in many countries, including Poland [48].

Transmission of antimicrobial-resistant *Campylobacter* strains to humans usually occurs via consumption of contaminated food products, especially of poultry origin [49]. Interestingly, high levels of resistance to tetracyclines among clinical isolates have often been observed despite the fact that these antibiotics are not used for campylobacteriosis treatment in humans in Europe [24]. However, antibiotics of this class are broadly applied in veterinary medicine for food-producing animals, which are an important reservoir of *Campylobacter* tetracycline-resistant strains [47,50]. On the other hand, the common use of antibiotics in veterinary medicine does not always correlate with AMR of *Campylobacter* recovered from animals treated in this manner [28]. Although the use of tetracyclines and macrolides has decreased in recent years, the percentages of these antimicrobial-resistant strains have remained stable or even increased [51]. On the other hand, the reverse trend has been noted for fluoroquinolones.

Based on antimicrobial resistance data, especially regarding the high prevalence of fluoroquinolone, tetracycline, and β-lactam resistance, as well as the lower but increasing resistance to macrolides, constant monitoring of the susceptibility of *Campylobacter* to antimicrobial agents is needed. Such measures, especially related to food-producing animals such as poultry, have been implemented in European Union member states since 2014 [52]. This will help to reduce antimicrobial-resistant *Campylobacter* strains among the main reservoirs of these bacteria and prevent the spread of resistance genes between animals as well as between animals and humans.

**Table 1 antibiotics-11-01079-t001:** Examples of prevalence of antimicrobial-resistant *Campylobacter* spp. isolated from various sources.

Resistance to Antimicrobials (Resistance Genes)	Source of Isolates	Percentage of Isolates with Resistance Genes	References
Fluoroquinolones (*gyrA*)	Humans	13.8 (Burkina Faso); 20.1 (Australia); 50.0 (Ethiopia); 55.8–85.7 (BE); 72.2–100 (Lithuania); 77.4 (Peru); 85.2 (PL); 89.4 (China)	Sangaré et al. [53]; Chala et al. [54]; Meistere et al. [49]; Wieczorek et al. [55]; Elhadidy et al. [56]; Zhang et al. [57]; Wallace et al. [58]; Schiaffino et al. [59]
	Animals and food	0–60.0 (China); 3.7–8.0 (USA); 25.0–100 (Ethiopia); 36.4–100 (Tunisia); 47.6–100 (Lithuania); 60.0–96.1 (Poland); 65.0–95.3 (Germany); 71.0 (Kenya)	Tang et al. [60]; Chala et al. [54]; Béjaoui et al. [61]; Nguyen et al. [62]; Meistere et al. [49]; Tenhagen et al. [50]; Wieczorek and Osek [48]; Andrzejewska et al. [63]; Bailey et al. [64]
Macrolides (*ermB*; efflux pumps)	Humans	0 (Lithuania); 0.6 (Poland); 1.8 (Australia); 2.0–28.6 (Belgium); 5.3 (Peru); 10.3 (Burkina Faso); 24.0 (China); 80.0 (Ethiopia)	Sangaré et al. [53]; Chala et al. [54]; Meistere et al. [49]; Wieczorek et al. [55]; Elhadidy et al. [56]; Zhang et al. [57]; Wallace et al. [58]; Schiaffino et al. [59]
	Animals and food	0–1.4 (Lithuania); 0–70.0 (China); 0–74.4 (Poland); 0–64.5 (Germany); 1.5–2.8 (USA); 25.0–100 (Ethiopia); 25.8–51.6 (Kenya); 90.9–100 (Tunisia)	Tang et al. [60]; Chala et al. [54]; Béjaoui et al. [61]; Nguyen et al. [62]; Meistere et al. [49]; Tenhagen et al. [50]; Wieczorek et al. [48]; Andrzejewska et al. [63]; Bailey et al. [64]
Tetracyclines (*tet(O)*)	Humans	10.3 (Burkina Faso); 15.6 (Australia); 49.7–85.7 (Belgium); 55.5–100 (Lithuania); 55.8 (Peru); 70.0 (Ethiopia); 70.3 (Poland); 93.3 (China)	Sangaré et al. [53]; Chala et al. [54]; Meistere et al. [49]; Wieczorek et al. [55]; Zhang et al. [57]; Wallace et al. [58]; Schiaffino et al. [59]
	Animals and food	0–64.0 (China); 3.8–77.0 (Poland);14.0–66.7 (Lithuania); 32.5–92.1 (Germany); 55.6–100 (Ethiopia); 57.9–78.1 (Poland); 65.3–81.6 (USA); 71.0 (Kenya); 100 (Tunisia)	Tang et al. [60]; Chala et al. [54]; Béjaoui et al. [61]; Nguyen et al. [62]; Meistere et al. [49]; Tenhagen et al. [50]; Wieczorek et al. [53]; Andrzejewska et al. [63]; Bailey et al. [64]

### 3.2. Enterococcus spp.

Bacteria of the genus *Enterococcus* are part of the natural gut microbiota in human and animals.

In human *Enterococcus* spp. are generally associated with incidences of intra-abdominal infections (IAI) and represent the second most common cause of infection. The *Enterococcus* bacteria are isolated from human with endocarditis, bloodstream infection, wound and surgical-site infection, as well as intra-abdominal and urinary tract infection [65]. This bacteria has also been associated with chronic periodontitis and persistent endodontic infections [66]. Typically, uncomplicated wound infections, urinary tract infections and most intra-abdominal infections are treated with a single antibiotic directed against enterococci.

Infections caused by *Enterococcus* spp. represent a serious threat to human and animal health due to difficulties in treatment. These bacteria are a very able to transfer of antimicrobial resistance genes and for this reason they are often resistant to many antimicrobials [67].

In clinical practice, combination therapy with a cell wall-activating agent (ampicillin, penicillin) and a synergistic aminoglycoside (high doses of gentamicin) is undertaken to treat serious enterococcal infections in critically ill patients and those with evidence of sepsis, as well as patients with endocarditis, meningitis, osteoarthritis. Once there is resistance to one antimicrobial from these groups of antibiotics, the drug of so-called last resort is vancomycin. For infections with vancomycin-resistant *Enterococcus* strains, linezolid, daptomycin, quinupristin/dalphopristin and tigecycline are used for treatment [68].

In veterinary medicine, the therapeutic management of *Enterococcus* spp. infections is mainly based on monotherapy with antibiotics selected based on the results of drug susceptibility testing, the location of the infection, and the species of affected animals [67,69]. This is related to the limited possibilities of using antibiotics in animals, especially those intended for consumption [67,70]. Such treatment appears to be effective, although randomized, controlled trials evaluating the efficacy of such treatment are lacking.

The last decade has seen more frequent reports of the role of *Enterococcus* spp. in the pathology of bird diseases. They have natural resistance to a number of antibiotics used in treatment, including β-lactam antibiotics, all generations of cephalosporins and sulphonamides. They show lower resistance to aminoglycosides, lincosamides and quinolones [65] Irrational use of antimicrobials has led to an increase in the population of multi-drug resistant *Enterococcus* strains.

#### 3.2.1. Resistance to β-Lactam Antibiotics

Enterococci have natural, innate resistance to β-lactam antibiotics due to their low affinity for penicillin-binding proteins—PBP5 in *E. faecium* and PBP4 in *E. faecalis* [71]. This resistance varies depending on the type of β-lactams, e.g., penicillin has the highest activity against enterococci, that of carbapenems is somewhat lower, and the activity of cephalosporins is the lowest. Another resistance mechanism associated with penicillin-binding proteins is sometimes observed in the case of bacteria which acquire a very high level of resistance to β-lactam antibiotics in comparison to wild strains. In some bacteria, resistance to β-lactams is associated with overproduction of PBP5 surface proteins. An example is overproduction of PBP5 by some penicillin-resistant strains of *E. hirae*. The gene *pbp5* in *E. hirae* is under the control of the *psr* gene [72]. Inactivation of *psr* through deletion or mutation leads to an increase in the number of copies of PBP5, and thus to saturation of all molecules of this protein. Some enterococci have a completely different, less common mechanism of resistance to β-lactams, involving synthesis of β-lactamases—enzymes hydrolysing the β-lactam ring in the antibiotic molecule. The hydrolysed antibiotic is inactivated and does not inhibit the enzymatic functions of surface PBPs. The gene determining expression of β-lactamases is located on a plasmid and usually occurs with a gene encoding resistance to gentamicin. Β-lactamases are most often produced in small amounts. Thus, when the number of bacteria is small, the MIC values for penicillin and ampicillin may be at a level corresponding to that of bacteria that are susceptible to these antibiotics [73].

#### 3.2.2. Resistance to Inhibitors of the Third Step of Peptidoglycan Synthesis—Glycopeptides

The most important process providing bacteria with resistance to glycopeptides is weakening of the bonds between molecules of the antibiotic and receptors located in the cell wall. Enterococci, both susceptible and resistant to vancomycin, possess specific complexes composed of peptidoglycans and pentapeptides outside their cell membrane. These are target structures to which glycopeptide antibiotics bind. The above-mentioned pentapeptides consist of tripeptide precursors to which dipeptides attach. In the case of vancomycin-susceptible strains, this is D-alanyl-D-alanine (D-Ala-D-Ala), while in the case of vancomycin-resistant enterococci (VRE) the terminal amino acid D-alanine is replaced with D-serine (D-ser) or D-lactate (D-lac). The D-alanine to D-serine substitution results in changes in the conformation of bonds with molecules of the antibiotic, significantly weakening it. If D-lactate is attached in place of D-alanine, the number of bonds with vancomycin decreases. Strains containing D-Ala-D-Ser dipeptides are susceptible to teicoplanin and show resistance to low concentrations of vancomycin. However, when peptidoglycan precursors contain D-Ala-D-Lac dipeptides, they are highly resistant to vancomycin [74]. Structures organized in this manner take part in cell wall synthesis. Due to binding of vancomycin to the pentapeptide molecule, it is blocked, stopping cell wall synthesis. In the second case, binding of the antibiotic to the peptidoglycan precursor is limited, and cell wall synthesis may take place despite the presence of vancomycin. Enterococci are neither phenotypically nor genotypically homogeneous in terms of resistance to vancomycin. Based on the level of resistance to vancomycin, the capacity to induce it, and cross-resistance to vancomycin and teicoplanin, seven phenotypic classes of enterococci are distinguished: VanA, VanB, VanC, VanD, VanE, VanG [75] and VanL [76]. Within each of the types VanB and VanC, three subtypes are distinguished (VanB1, VanB2, VanB3; VanC1, VanC2, VanC3). The most important phenotypes in clinical practice are VanA and VanB.

Type A resistance (VanA). This acquired, inducible resistance to high concentrations of vancomycin (MIC 64–100 μg/mL) and teicoplanin (MIC 16–512 μg/mL) is most common in *E. faecalis*, *E. faecium*, *E. durans*, *E. raffinosus*, *E. hirae*, *E. avium*, *E. casseliflavus*, and *E. mundtii*, but also occurs in *E. gallinarum* [77,78]. The terminal fragment of D-Ala-D-Ala is replaced with dipeptide D-Ala-D-Lac, which prevents binding of the antibiotic even at very high concentrations. Genes are transferred by transposon *Tn1546* present on a plasmid or chromosome [75].

Type B resistance (VanB). The VanB phenotype is characterized by inducible resistance to vancomycin at various levels (MIC 4–1024 μg/mL) with susceptibility in vitro to teicoplanin (MIC 0.5–1 μg/mL). Dipeptide D-Ala-D-Ala is replaced with D-Ala-D-Lac, as in phenotype VanA. It is found in the species *E. faecium*, *E. faecalis*, *E. durans* and *E. gallinarum* [79,80]. Based on DNA sequences of genes encoding VanB ligase, three subtypes are distinguished within this phenotype: *VanB1*, *VanB2*, and *VanB3* [77]. However, no relationship has been shown between a given subtype and the level of resistance to vancomycin and teicoplanin.

Type C resistance (VanC). This is natural resistance occurring in motile species of enterococci: *E. gallinarum* (*vanC1*) [79], *E. casseliflavus* (*vanC2*) and *E. flavescens* (*vanC3*) [81]. It is both inducible and constitutive. Alongside D-Ala-D-Ala fragments, D-Ala-D-Ser peptides appear in a 1:3 ratio. This phenotype is characterized by resistance to low concentrations of vancomycin (MIC 4–32 mg/L) and susceptibility to teicoplanin (MIC < 1 mg/L).

Type D resistance (VanD) results from production of peptidoglycan precursors terminating in D-alanyl-D-lactate, but the genetic background of strains described thus far is varied [82]. In the case of *Enterococcus faecium* BM439, insertion of five base pairs has been found in the structural gene of ligase, while in *E. faecium* BM4416 the mutation was due to insertion IS19 in the gene *ddl*. *VanD* enterococci have constitutive resistance to relatively high concentrations of vancomycin (MIC 64 µg/mL) and to low concentrations of teicoplanin (MIC 4 µg/mL).

Type E, G and L resistance (VanE, VanG and Van L). VanE is an acquired type of resistance found in *E. faecalis* strain BM4405, showing a low level of resistance to vancomycin (MIC 16 µg/mL) and sensitivity to teicoplanin. This strain produces peptidoglycan precursors terminating in D-alanyl-D-serine [83]. The acquired VanG phenotype is characterized by resistance to vancomycin (MIC 16 mg/mL) and susceptibility to teicoplanin (MIC 0.5 mg/mL). This resistance results from production of peptidoglycan precursors terminating in D-ala-D-ser [84]. The VanL phenotype has been found in *E. faecalis* N06-0364 resistant to vancomycin at a level of MIC 8 µg/mL. This resistance results from production of peptidoglycan precursors terminating in D-ala-D-ser. The *vanL* gene is similar in structure to the *vanC* operon, but VanT serine racemase is encoded by two separate genes: *vanTm*L (membrane-binding) and *vanTr*L (racemase) [76]. The VanD, VanE, VanG and VanL phenotypes are extremely rare. No clinical strains with these resistance phenotypes have been recorded in Poland.

#### 3.2.3. Resistance to Aminoglycosides

Bacterial resistance to aminoglycoside antibiotics is associated with the activity of enzymes that inactivate the antibiotic, enzymatic modification of 16S rRNA, a change in the permeability of bacterial membranes for the drug, or enzymatic modification of the antibiotic. The natural mechanism of resistance to low concentrations of aminoglycosides in enterococci is associated with low permeability of bacterial cell membranes for molecules of the antibiotic and precludes the use of these drugs in monotherapy. However, the combined use of an aminoglycoside with penicillin or glycopeptides is effective, provided the bacteria are susceptible in vitro to these groups of antibiotics [79]. Resistance to high concentrations of aminoglycosides—the HLAR (high-level aminoglycoside resistance) phenotype—is an acquired resistance. In these cases, bacteria are able to synthesize specific aminoglycoside-modifying enzymes (AME), which include phosphotransferases (APHs), acetyltransferases (AACs) and nucleotidyltransferases (ANTs). In the acylation and phosphorylation processes, enzymes modify molecules of the antibiotic, giving it a different conformation that prevents the drug from binding to its target site (30S ribosomal subunit). Expression of AMEs depends on genes located on mobile genetic elements—plasmids, which can easily spread among various strains [85]. Currently over 60 enzymes able to modify aminoglycoside molecules have been identified. However, there are *Enterococcus* strains that are resistant to aminoglycosides, which produce the bifunctional enzyme AAC(6′)-APH(2″), encoded by the gene *aac(6′)-Ie-aph(2″)-Ia*. This protein has a broad spectrum of activity, because it can modify many aminoglycoside antibiotics, except for streptomycin. *aac(6′)-Ie-aph(2″)-Ia* mainly mediates resistance to gentamicin, tobramycin, amikacin, kanamycin, netilmicin and dibekacin. More than 90% of *Enterococcus* strains isolated from clinical cases show a high level of resistance to gentamicin (MIC ≥ 2000 µg/mL), due to the presence of *aac(6′)-Ie-aph(2″)-Ia*, while less than 10% possess *aph(2″)-Ic*, *aph(2″)-Id* or *aph(2″)-Ib* [86].

The HLSR (high-level streptomycin resistance) phenotype means a high level of resistance to streptomycin -MIC > 1024 µg/mL EUCAST [87], which results from the activity of nucleotidyltransferase ANT(6′) or ANT(3″), modifying streptomycin, or to mutations in the genes encoding components of 30S ribosomal subunit. Due to mutations of genes responsible for ribosomal proteins, a new ribosomal protein is produced. The mutated protein S12, which is an aminoglycoside receptor, has very low affinity for this class of antibiotics. In this way *Enterococcus* bacteria with mutated ribosomal protein S12 acquire resistance to streptomycin.

#### 3.2.4. Resistance to Tetracyclines

Tetracyclines inhibit protein synthesis (by blocking the bacterial 30S ribosomal subunit) and disturb energy processes in bacterial cells. Antibiotics can be actively removed from the cytoplasm by special efflux pumps resulting from the expression of genes *tet*(K) and *tet*(L) and ribosome conformation. Genes *tet*(M), *tet*(O) and *tet*(S) encode proteins which affect resistance by protecting the ribosome. This involves binding resistance proteins to the ribosome, followed by a change in the conformation of the ribosome, which limits binding of tetracyclines. The most common gene of resistance to tetracyclines is *tet*(M), which is located on the chromosome and is usually transferred on transposon *Tn916* or similar conjugative transposons, but sometimes on conjugative plasmids [88].

#### 3.2.5. Resistance to Macrolides, Lincosamides and Streptogramins

The first described mechanism of resistance to macrolides involved post-transcription modification of 23S rRNA resulting from the activity of adenine-N^6^-methyltransferase. Enzymes from this group transfer one or two methyl residues to an adenine molecule in 23S rRNA, resulting in N^6^-methyladenine or N^6^,N^6^-dimethyladenine. This reduces binding not only of erythromycin in the ribosome, but also of other macrolides (azithromycin and clarithromycin) and of lincosamides and streptogramin B. This mechanism is encoded by the gene *erm*(B), or less often by *erm*(A), and the phenotype is called ‘MLS_B_’ (macrolide-lincosamide-streptogramin B) [89]. Resistance to macrolides and lincosamides can also be determined by the ability of bacteria to produce specific enzymes inactivating antibiotics. This inactivation can be the result of cleavage of the lactone ring of the macrolide by esterase or modification of its structure by enzymes with transferase activity (acetyltransferase, nucleotidyltransferase, or phosphotransferase), phosphorylase or hydrolase. These mechanisms usually give the bacterial cell resistance to only one of three classes of antibiotics belonging to this group (macrolides, lincosamides, or streptogramins), or to only one kind of them (e.g., streptogramin B). Some *Enterococcus* spp. bacteria have the *sat* gene encoding acetyltransferases which inactivate group A streptogramins. *E. faecalis* is naturally resistant to clindamycin (lincosamides), quinupristin (streptogramin B) and dalfopristin (streptogramin A), due to expression of the gene *lsa*. This gene is structurally similar to the ABC (ATP-binding cassette) transporters, suggesting ATP-energized efflux as the likely resistance mechanism against clindamycin and dalfopristin. The gene was found in 180/180 strains of *E. faecalis* and in none of 189 other enterococci, which suggests that it is innate in *E. faecalis* [90].

#### 3.2.6. Resistance to Antimetabolites—Sulphonamides and Trimethoprim

Sulphonamides inhibit synthesis of dihydrofolic acid, acting as competitive inhibitors of the enzyme dihydropteroate synthase (DHPS). Production of dihydropteroate synthase by bacteria causes their affinity for the drug to decrease. In addition, overproduction of p-aminobenzoic acid causes it to compete directly for access to the active centre of DHPS. The action of the antibiotic causes some bacteria to produce an alternative metabolic pathway, which replaces the pathway blocked by the antibiotic. This phenomenon is called a ‘bypass mechanism’ and is recognized as a means of acquiring resistance to sulphonamides and trimethoprim. Most bacteria are not able to absorb folacin from the environment, so they need to synthesize it to produce nucleic acids. The combination of trimethoprim and sulfamethoxazole inhibits two consecutive steps in the tetrahydrofolate synthesis pathway, which blocks synthesis of folic acid and synergistically destroys a broad spectrum of bacteria. *Enterococcus* bacteria have the exceptional ability to absorb folic acid from the environment, thereby bypassing the effect of the trimethoprim/sulfamethoxazole combination [91].

#### 3.2.7. Resistance to Fluoroquinolones

Quinolones inhibit bacteria by interacting with type II topoisomerase, DNA gyrase and topoisomerase IV, which are essential to replication of bacterial DNA. DNA gyrase consists of two subunits called *gyrA* and *gyrB*. Topoisomerase IV, which is the main target of quinolones in gram-positive bacteria, consists of two subunits called *parC* and *par E*, which are homologous to *gyrA* and *gyr B*. Resistance of enterococci to fluoroquinolones is determined by point mutations on chromosomes. Therefore, genes of resistance to fluoroquinolones cannot be transferred to other bacteria via transfer of genetic material. The frequency of this mutation in the bacterial genome is determined by the intensity of antibiotic treatment with drugs from this group. The most common mutations are modifications of genes encoding topoisomerase II (gyrase) and topoisomerase IV. *E. faecalis* with a mutation in *parC* but not in *gyrA* has shown intermediate resistance to quinolones. The minimum inhibitory concentration (MIC) of quinolones for this isolate was higher than the MIC for *E. faecalis* without a mutation in the *parC* or *gyrA* gene, but lower than the MIC for *E. faecalis* with mutations in both *parC* and *gyrA* [92].

Among 911 tested *Enterococcus* spp. isolates (mainly *E. faecalis* and *E. faecium*) from poultry, the highest resistance was shown for trimethoprim/sulfamethoxazole (88%), tylosin (71.4%), enrofloxacin (69.4%), doxycycline (67.3%) and lincomycin with spectinomycin (56.1%). Enterococci isolated from wild birds showed the highest resistance to lincomycin (100%), tetracycline (48%), erythromycin (44%) and ciprofloxacin (22%). These birds also showed resistance to high concentrations of streptomycin and kanamycin (in 19% and 15% of isolates, respectively) [93]. Another study [94], analysing the drug susceptibility of 227 enterococci isolated from animals, fresh food, hospital and municipal wastewater, and seawater in Poland showed the highest susceptibility to penicillin, ampicillin, vancomycin, teicoplanin, tigecycline, linezolid and daptomycin. Some of these isolates showed resistance to high concentrations of gentamicin (1.3%) and/or streptomycin (9.2%) and were often not susceptible to rifampin and tetracycline (81.9% and 53.7%, respectively). Enterococci isolated from sick dogs (urinary tract infections in own study) Stępień-Pyśniak et al. [69] and healthy dogs (gastrointestinal tract) in Italy showed a high level of resistance to aminoglycosides (streptomycin 94.1%, neomycin 90.2%, gentamicin 68.6%), fluoroquinolones (enrofloxacin 74.5%, ciprofloxacin 66.7%), oxacillin (98%), clindamycin (84.3%), tetracycline (78.4%) and quinupristin-dalfopristin (78.4%) [69]. Clinical *Enterococcus* spp. isolates from humans in Italy showed high resistance to ampicillin (84.5%), ampicillin/sulbactam (82.7%) and imipenem (86.7%) as well as to high-level gentamicin and streptomycin [95].

### 3.3. Escherichia coli

*Escherichia coli* (*E. coli*) is a common member of the natural intestinal microflora of humans and animals. Within the *E. coli* species, apart from commensals commonly colonizing the intestines of mammals and birds, there are also intestinal pathogenic *E. coli* (IPEC) and extraintestinal pathogenic *E. coli* (ExPEC) strains. IPEC bacteria are associated with infections of the gastrointestinal tract. Several pathotypes can be distinguished among IPEC strains: enteropathogenic *E. coli* (EPEC), enterotoxigenic *E. coli* (ETEC), enterohaemorrhagic *E. coli* (EHEC), enteroinvasive *E. coli* (EIEC), enteroaggregative *E. coli* (EAEC), adherent invasive *E. coli* (AIEC), and diffusely adherent *E. coli* (DAEC) [96]. ExPEC strains cause infections in extraintestinal anatomic sites, and in this group include: uropathogenic *E. coli* (UPEC) associated with urinary tract infection in human and animals, neonatal meningitis-associated *E. coli* (NMEC), septicaemic *E. coli* (SePEC) causing systemic infection in human and animals, avian pathogenic *E. coli* (APEC) that cause avian colibacillosis, and a potentially emerging ExPEC lineage named endometrial pathogenic *E. coli* (EnPEC) [97,98].

*E. coli* infections in animals are subjected to various pharmaceutical treatments. For instance, broad-spectrum cephalosporins and fluoroquinolones are commonly used to treat bovine mastitis [99]. One of the antibiotics used to treat colibacillosis in poultry is enrofloxacin, which belongs to fluoroquinolones [100]. Antibiotics belong to β-lactams, polymyxins, tetracyclines and macrolides have been reported to be the most frequently used antibiotics in European pig production, mainly used to treat intestinal and respiratory disorders [101].

One of the most important mechanisms of resistance to antibiotics in *Enterobacteriaceae* is the production of enzymes capable of hydrolysis of penicillin, cephalosporin and monobactams. These enzymes are called extended-spectrum β-lactamases (ESBLs). The mechanism was detected in 1983 in *Klebsiella pneumoniae*. Initially, bacteria with ESBLs were known only as aetiological agents in hospital-acquired infections. They are currently also detected as a cause of community-acquired infections, and cases of carriage of these bacteria are noted as well.

The mechanism, which was first diagnosed in 2008 *in K. pneumoniae* and *E. coli*, involves production of an enzyme from the metallo-β-lactamase group. It confers resistance to many β-lactam antibiotics, including carbapenems. Carbapenems are regarded as drugs of last resort and are used mainly to treat diseases caused by bacteria resistant to other antibiotics. Substances from this group are especially important due to their broad spectrum of activity. Carbapenems exhibit activity against aerobic and anaerobic, gram-positive, and gram-negative bacteria [102].

#### 3.3.1. Resistance to β-Lactams

Production of β-lactamases by bacteria can be induced or constitutive. Constitutive enzymes are continually produced by cells, and their level is not dependent on the presence of an antibiotic in the environment but is a fixed trait of the strain or species of bacteria. The presence of constitutive β-lactamase, produced in an adequate amount, can be the cause of natural resistance to an antibiotic to which it has affinity. The condition of production of induced β-lactamases by bacteria is activation by the antibiotic of β-lactam present in the environment. Synthesis of these enzymes takes place until the activator is removed from the environment, and thus it determines temporary resistance. Mutations may occur during induction, so that the induced β-lactamase is continually produced by the bacteria, even after the inductor is removed, resulting in permanent resistance to the antibiotic [103].

Resistance of *E. coli* to β-lactams is mainly associated with the production of extended-spectrum β-lactamases (ESBL), which are capable of hydrolysing penicillins, cephalosporins (except for cephamycin) and monobactams, but are not able to break down carbapenems, and their activity is inhibited by β-lactamase inhibitors (clavulanic acid, sulbactam, and tazobactam) [104]. Β-lactamases are present in the periplasmic space, most often encoded by large plasmids (e.g., IncF, IncI1, IncN, IncHI1, and IncHI2 [105]), which facilitates their rapid and uncontrolled spread. Β-lactamases are a highly diverse group of enzymes in terms of structure and substrate profile, i.e., activity against penicillin, cephalosporins and carbapenems [106]. Several types of ESBLs are distinguished: CTX-M, SHV, TEM, OXA, PER, VEB, BES, GES, SFO, TLA and IBC [107]. The enzyme CTX-M exhibits strong hydrolytic activity against cefotaxime, encoded by the gene *bla*_CTX-M_, usually located on a plasmid e.g., IncF, IncN, INcK and Incl1 [105]. OXA enzymes, which hydrolyse carbapenems, are mainly found in Enterobacterales (including *E. coli*) and are coded by *bla*_OXA_ plasmid genes. ESBLs are most widespread among Enterobacteriaceae, including *E. coli*. Apart from those produced by hospital-acquired strains, they are increasingly observed in strains inducing community-acquired infections, as well as in bacterial strains isolated from animals [108]. It should be noted that in many *E. coli* isolates of animal origin, a common location was identified for the *mcr*-1 gene determining resistance to colistin and *bla* genes encoding extended-spectrum β-lactamases.

##### AmpC β-Lactamases

AmpC cephalosporins break down all penicillins and most cephalosporins (mainly first and second generation) except for cefepime, but do not hydrolyse aztreonam, although some of them can bind it [109]. These enzymes do not hydrolyse carbapenems and are not inhibited by clavulanic acid. They are assigned to class C according to the Ambler classification. Expression of this type of β-lactamase is usually inducible, but in the case Of *E. coli* it takes place constitutively at a very low level and does not confer β-lactam resistance to wild strains [110]. Chromosomally encoded AmpC β-lactamases are identified among *E. coli* strains from both humans [111] and animals [112].

#### 3.3.2. Resistance to Fluoroquinolones

In *E. coli*, mutations associated with an increase in resistance to fluoroquinolones are most often observed in the protein GyrA and in ParC, in which a change in the encoded amino acid alters the properties of the protein. The amino acids most often undergoing substitution in GyrA in *E. coli* are Ser83 and Asp87 [113]. It should be noted that single mutations in the *gyr*A gene can contribute to the resistance of bacterial strains to quinolones, but in order for resistance to develop, additional mutations are required in *gyr*A and/or *par*C. In clinical *E. coli* isolates 1–4 mutations have been detected in *gyr*A, *gyr*B and *par*C from strains isolated from dogs and cats [114]. In addition, there are mechanisms of resistance to fluoroquinolones which are encoded by plasmids—plasmid-mediated quinolone resistance (PMQR). For plasmids carrying PMQR genes belong RCR, IncF, IncN, IncX1, IncX2. [105]. Clinically they are less important than mechanisms of mutation at the target site, i.e., gyrase and topoisomerase. This group of mechanisms includes Qnr proteins changing the target site of the antibiotic and structurally modifying fluoroquinolones and aminoglycosides, a bifunctional variant of the enzyme aminoglycoside acetyltransferase—AAC(6′)-Ib-cr, and efflux pump proteins associated with active transport—QepA and OqxAB [115]. These determinants do not lead to a high level of resistance to quinolones/fluoroquinolones, only reducing susceptibility to them. Nevertheless, they can increase resistance to this group of antimicrobials by co-existing with mechanisms of resistance encoded chromosomally. All these mechanisms of resistance to fluoroquinolones have been confirmed in *E. coli* isolated from companion animals, livestock and humans [116,117].

Another mechanism of resistance, involving active removal of fluoroquinolone molecules from the bacterial cell in order to reduce the concentration of the drug in the cytoplasm, is complexes of transport proteins called efflux pumps [118]. In the case of *E. coli*, several pump systems have been identified (AcrAB-TolC, EmrAB, MdfA, TehA, EmrE, AcrE, and EmrD). The best understood are mechanisms associated with the presence of AcrAB-TolC efflux pumps [119].

#### 3.3.3. Resistance to Aminoglycosides

One cause of resistance to aminoglycosides is chromosomal mutations altering the binding sites of the antibiotic and modification of 16S rRNA mediated by methylases. Actinobacteria with the ability to produces aminoglycosides protect themselves against their own aminoglycoside metabolites by producing 16S ribosomal RNA methyltransferase (16S-RMTase), which prevents them from binding to 16S rRNA. Ten 16S-RMTases have thus far been identified: ArmA, RmtA, RmtB, RmtC, RmtD, RmtE, RmtF, RmtG, RmtH and NmpA [120]. In *E. coli*, genes coding for 16S-RMTase, *armA, rmtB, rmtD*, and *rmtE* have been detected in strains from chickens, pigs and cattle [121,122]. Only one acquired 16S-RMTase—NpmA—has been detected in *E. coli*. This enzyme was identified in a clinical human strain of *E. coli* in Japan in 2007 [123]. NpmA is responsible for resistance to gentamicin, tobramycin, and amikacin, but also confers a broader spectrum of resistance to this group of antibiotics, including neomycin and apramycin. It should be noted that some genes encoding 16S-RMTases co-exist with other factors responsible for resistance to antibiotics. An example is the high prevalence of the gene *arm*A in *Enterobacteriaceae* producing NDM (New Delhi metallo-β-lactamase) carbapenemase, which is also produced by *E. coli*. Studies of plasmids transferring *bla*NDM have shown that the gene often co-exists with the *arm*A gene or other 16S-RMTase genes (especially *rmt*B, *rmt*C and *rmt*F) on the same plasmids [120]. Pulss et al. [124] identified an *E. coli* isolate from pigs which in addition to *arm*A contained other genes: *bla*_CMY-2_, *bla*_OXA-181_ and *mcr*-1.

Another mechanism of resistance to aminoglycosides is enzymes modifying aminoglycoside antibiotics. These can be divided into three groups depending on their mechanism of action: acetyltransferases (AAC), which catalyse acetylation of the amine group of amino sugars, nucleotidyltransferases (ANT), which attach nucleotide molecules from ATP to the hydroxyl group of amino sugars located in the aminoglycoside molecule, and phosphotransferases (APH), responsible for phosphorylation of the hydroxyl group of the sugar residue of the antibiotic. AAC, ANT and APH can be located on a plasmid, chromosome or integron. Several dozen AACs inducing resistance to aminoglycosides have been identified in various genera of bacteria, many of which have been confirmed in *E. coli*. In the case of *E. coli* strains from animals and humans, the most commonly identified are acetyltransferases AAC(3)-II/IV and AAC(6)-Ib [125,126]. Among aminoglycoside phosphotransferases characteristic of *E. coli* isolated from various animal species, the most common are APH(6)-Ia and APH(6)-Id, encoded by *str*A and *str*B, respectively. Examples of nucleotidyltransferases found for *E. coli* include ANT(2″) encoded by *aad*B and ANT(3″) encoded by *aad*A [127].

#### 3.3.4. Resistance to Tetracyclines

The mechanism of resistance to tetracyclines most often involves efflux pumps, tasked with pumping the antibiotic out of the cytoplasm. Genes encoding efflux systems which determine resistance to tetracyclines in *E. coli* include *tet(A), tet(B), tet(C), tet(D), tet(E), tet(G), tet(J), tet(L)* and *tet(Y)*. Another mechanism of resistance to tetracyclines is proteins whose task is to protect the ribosome. By binding to ribosome they limit binding of tetracyclines to it. In *E. coli* this group includes proteins encoded by two genes: *tet*(M) and *tet(W)*. In addition, the gene *tet(X)* has been identified for *E. coli*; it encodes oxidoreductase, whose function is inactivation of first- and second-generation tetracyclines [128]. The most commonly recorded genes encoding efflux systems are *tet(A), tet(B)* and indirectly *tet(C)* [129,130]. It should be noted that in bacteria which have been tested for the presence of *tet* genes, they have not always been identified singly. For example, Jahantigh et al. [131], in addition to strains carrying a single *tet* gene mainly *tet(A)* and also *tet(B), tet(C)* and *tet(D)*, confirmed the presence of isolates carrying two or three *tet* genes simultaneously (43.3% and 13.3%, respectively).

#### 3.3.5. Resistance to Sulphonamides and Trimethoprim

Mechanisms determining resistance to sulphonamides mainly include the presence of *sul* genes encoding dihydropteroate synthase with low affinity for sulphonamides, which means that the bacteria replicate normally in an environment containing sulphonamides. Currently four genes of resistance to sulphonamides (*sul1*, *sul2*, *sul3* and *sul4*) have been identified on plasmids [132]. IncFII and IncI1 are plasmids carrying genes encoding resistance to suplhonamides [133]. Resistance to trimethoprim is determined by the presence of the *dfr* gene encoding dihydrofolate reductases, which are not susceptible to this antimicrobial. The *sul1* gene is located in the gene cassette in the variable part of class 1 integrons and often co-exists with other resistance genes. Genes *sul2* and *sul3* are also located on plasmids on which other determinants of resistance are present. The most commonly recorded genes determining resistance to sulphonamides in *E. coli* are *sul*1 and *sul*2 [134,135].

The *dfr* genes determining resistance to trimethoprim have been identified many times in various gram-negative bacteria, including *E. coli*. They have been divided into two groups based on their size and structure: *dfrA* and *dfrB*. The *dfrA* genes encode proteins 152 to 189 amino acids in length, whereas proteins encoded by another group of genes—*dfrB*—have only 78 amino acids. Both genes are located on gene cassettes, which are insIrted into class I or II integrons [136]. In the case of *E. coli*, the vast majority of genes identified are *dfrA* genes [137].

#### 3.3.6. Resistance to Phenicols

Due to the high toxicity and numerous side effects of nonfluorinated phenicols, chloramphenicol and its derivatives are no longer used in treatment of animals intended for consumption. They are still used, however, in treatment of pets. The first mechanism of bacterial resistance to chloramphenicol, which remains the most common, is enzymatic inactivation through acetylation of the drug by various types of chloramphenicol acetyltransferases—CATs (the mechanism is manifested as the presence of the *cat* gene). Chloramphenicol acetyltransferase modifies the antibiotic by transforming it into inactive derivatives—monoacetates or diacetates. The chloramphenicol acetyltransferase gene is usually encoded on a plasmid or transposon and can transpose to the chromosome [138]. Expression of the enzyme in the case of *E. coli* is encoded constitutively. Other mechanisms of resistance include active efflux of nonfluorinated phenicols (presence of the gene *cmlA*) or fluorinated and nonfluorinated phenicols (presence of the gene *floR*) from the bacterial cell and the activity of rRNA methylase encoded by the gene *cfr* [139]. In *E. coli* isolated from animals, CATs belonging to group A1 (*cat*I gene), B2 (*cat*B2 gene) and B3 (*cat*B3 gene) have been identified [140].

#### 3.3.7. Resistance to Polymyxins

Polymyxins influence both the outer and cytoplasmic membrane of various species of gram-negative bacteria, including *Enterobacteriaceae*, also acting on *E. coli*. One of the mechanisms of resistance to polymyxins in *E. coli* strains is modification of lipid A of LPS through the addition of phosphoethanolamine (PEtN) and/or 4-amino-4-deoxy-L-arabinose (L-Ara4N), which decreases the affinity of colistin to the bacterial cell. This chromosomal resistance mechanism to colistin in *E. coli* is the result of activation of two-component systems PmrA/PmrB and PhoP/PhoQ by specific mutations of genes (*pmr*A, *pmr*B, *pho*P, and *pho*Q) encoding proteins of these systems or environmental stimuli leading to overexpression of genes modifying LPS [141]. The first gene of resistance to polymyxins was identified in 2015, located on a plasmid from an *E. coli* strain from a pig. The gene encoding phosphoethanolamine transferase MCR-1 was designated *mcr*-1 [142]. The following year a paper was published in which Xavier et al. [143] identified another gene located on a plasmid, *mcr*-2, determining colistin resistance, with over 77% similarity to *mcr*-1. Thus far several *mcr* genes and their variants have been isolated from *E. coli* strains of animal origin [144,145].

*E. coli* isolates tested in various parts of the world, obtained from human, animals and directly from the environment, have shown the highest resistance to amoxicillin (70.5–95%), while the smallest percentage of strains have been resistant to colistin (0.8%). By far the highest percentage of resistance to amoxicillin has been noted in strains isolated from animals [146]. High resistance was also observed for cefotaxime (~60%), ceftazidime (50–58%), tetracycline (50–55%) and ampicillin (45–50%). It should be noted that strains from humans and animals have shown similar resistance to these antibiotics. *E. coli* strains isolated from poultry in Poland have shown a high percentage of strains resistant to ampicillin (100%), doxycycline (100%), ciprofloxacin and streptomycin (81.3%), and amoxicillin/clavulanic acid (75%) [147].

The examples of the prevalence of antibiotic resistance gens in *E. coli* strains isolated from the environment and different animal species has been presented in Table 2.

### 3.4. Listeria spp.

Bacteria of the genus *Listeria* are the aetiological agent of listeriosis in humans and many animal species, including birds. The genus *Listeria* currently includes 17 species present throughout the environment, among which the most commonly isolated is *L. monocytogenes*, which is the third most common cause of death from food poisoning in human. *Listeria ivanovii* causes infections mainly in ruminants. Infections in humans caused by *L. monocytogenes* are especially common in risk groups such as pregnant women, infants, the elderly, and human with reduced immunity [166]. Listeriosis in human is usually treated with gentamicin in combination with and amoxicillin or ampicillin. Other drugs of choice for pregnant women are erythromycin, vancomycin, and trimethoprim/sulfamethoxazole [167].

Resistance of bacteria to antibiotics is associated with the presence of antibiotic resistance genes especially in moving fragments. It was also described the transfer of plasmids by conjugation with *Enterococcus* spp., *Staphylococcus* spp., *Streptococcus* spp. plasmids and transposons carrying antibiotic resistance genes from these bacrogram specis to *Listeria* and as well as between species of *Listeria* [168]. It has been shown that in Listeria spp., the main mechanism responsible for the development of antibiotic resistance is the acquisition of mobile genetic elements, e.g., self-transferable, mobilizable plasmids and conjugative transposons [168]. Such mechanisms, referred to as efflux pumps, are significantly associated with the occurrence of drug resistance in L. monocytogenes to fluoroquinolones, macrolides and cefotaximes.

#### 3.4.1. Resistance to Fluoroquinolones

In the case of resistance to fluoroquinolones, there are three plasma-mediated mechanisms of resistance (PMQR) as well as chromosomal resistance associated with mutations causing modifications of amino acid sequences of individual subunits of topoisomerase II and IV [169]:PMQR (plasmid-mediated quinolone resistance) mechanisms associated with the presence of Qrn proteins (Qrn A, B, S, and less often C and D), responsible for protecting bacterial DNA and the enzymes gyrase and topoisomerase IV. The bifunctionality of the variant of the enzyme aminoglycoside acetylotransferase (AAC6′)-lb-cr, through modification of the aminoglycoside molecule, leads to the loss of affinity of subunit 16S rRNA to drugs such as tobramycin, kanamycin or amikacin;Active removal of fluoroquinolones from the interior of the cell by efflux pump proteins QepA and OqxAB, e.g., through overexpression of the chromosomal genu *lde*, encoding pump proteins;Point mutations in QRDRs (quinolone resistance-determining regions) in the genes *gyrA* and *gyrB* encoding subunits of topoisomerase II (gyrase) or in the genes *parC* and *parE*, responsible for encoding subunits of topoisomerase IV, the second main target enzyme of fluoroquinolones besides topoisomerase II. It should be noted that both enzymes play a major role during replication, transcription, recombination, and repair of bacterial DNA.

PMQR mechanisms alone cause only a low level of resistance to fluoroquinolones, but they promote selection of mutations in the gyrase and topoisomerase IV genes and acquisition of high-level resistance to fluoroquinolones by bacteria. Numerous reports indicate that resistance to fluoroquinolones is a complex process resulting from the interaction of several mechanisms determining multi-drug resistance in bacteria, e.g., PMQR together with ESBL, *pAmpC*, and KPC and mutations in QRDRs in subunits of topoisomerase II and IV [169].

#### 3.4.2. Resistance to Macrolides

Macrolides antibacterial activity is with binding to the 50S ribosomal subunit which inhibit the biosynthesis of 23S ribosomal RNA (rRNA). In Listeria spp., the resistance against macrolides i.e., erythromycine is connected with the presence of two genes for resistance *ermB* and *ermC*. The erm (erythromycin ribosome methylase) genes encode methyltransferases that modify 23S rRNA [170].

#### 3.4.3. Resistance to Tetracyclines

In the case of *Listeria* spp., thus far five determinants of resistance to tetracyclines have been identified: *tet(K), tet(L), tet(M), tet(S)* and *tet(T)*. The *tet(M), tet(S)* and *tet(T)* genes are responsible for encoding specific cytoplasmic proteins protecting ribosomes against antibiotics, while the *tet(L)* and *tet(K)* genes encode outer membrane proteins, responsible for eliminating the antibiotic from the cell by active transport. Tetracycline resistance can be transferred conjugatively between the different bacterial strains i.e., *Enterococcus* spp. and *Listeria* spp. [171,172]. Resistance to a given antibiotic in different genera of bacteria may also be linked to the occurrence of the same gene, e.g., in the case of resistance to tetracycline and minocycline with the gene *tet(M)*, which is the most common not only in *Listeria* but also among other gram-positive bacteria of the genera *Enterococcus, Staphylococcus* and *Streptococcus*, as well as some gram-negative bacteria [172]. The *tet(M*) has also been shown to be linked to large integrative and conjugative elements (ICEs) with a broad range, e.g., Tn916 or Tn1545 [173].

### 3.5. Resistance to β-Lactams

An enzyme that determines resistance to beta-lactams as a result of hydrolysis in Gram-positive bacteria (including *Listeria* spp.) Ferro Β-lactamase, which may be located on the chromosome, plasmids or genetic material of bacterial phages [174] The presence of *fosA, fosB, fosX* genes has been confirmed in gram-negative (*P. aeruginosa*) and gram-positive (*S. aureus, L. monocytogenes*) bacteria.

Extensive results of research on the occurrence of genes of resistance in strains of *L. monocytogenes* isolated from humans and the environment were presented by Hanes and Huang [175]. The authors showed a high prevalence of the genes *fosX*, *lin*, *abc-f* and *tet(M)* as the four most commonly occurring genes in *L. monocytogenes* in Europe, Australia and New Zealand, South America, South Africa and UK/Ireland. Examples of the prevalence of genes of resistance in *Listeria* spp. isolated from various sources are presented in Table 3.

Some studies [176] have confirmed the occurrence of resistance to several groups of antibiotics in *Listeria* spp. For example, *L. monocytogenes* strains isolated from more than a dozen dairy cattle farms showed resistance to ampicillin (92%, MIC> or =2), rifampicin (84%, MIC > or =4), rifamycin (84%, MIC > or =4), florfenicol (66%, MIC> or =32), tetracycline (45%, MIC> or =16), penicillin G (40%, MIC> or =2) and chloramphenicol (32%, MIC> or =32).

A study by Kayode et al. [177] carried out in the United States confirmed high resistance to several groups of antimicrobials in *L. monocytogenes* strains isolated from the natural environment, surface water (rivers), wastewater, and irrigation water. Strains showed a high percentage of resistance to sulphonamides (sulfamethoxazole, STX) > 63.16%, tetracycline (oxytetracycline) 54.39%; β-lactams (amoxicillin) 50.88%, penicillin G 40.35%; aminoglycosides (streptomycin) 47.37%; cephalosporins (cefotetan) 45.61% and macrolides (erythromycin) 43.85%. The values for phenicols (chloramphenicol) and vancomycin were <40% (38.60% and 36.84%, respectively).

In the study Hailu et al. [178] the authors confirmed the 100% multi-resistance (MDR) of *L. monocytogenes* isolates obtained from dairy and poultry farms, manure, and soil in Ohio, USA to lincomycin (clindamycin), Rifamicines (ifampicin), cephalosporins (ceftriaxone, cefoxitin), Penem (meropenem), macrolides (azithromycin) and trimethoprim sulfamethoxazole, with 100% resistance for each antimicrobial class. A high resistsnce among the tested strains was also observed to Aminoglycosides- streptomycin (98.5%), Quinolones-nalidixic acid (95.5%), Fluoroquinolone—levofloxacin (91%) and penicillin- ampicillin (89.5%). As wrote Authors all *L. monocytogenes* strains isolated from poultry farms were resistant to kanamycine, nalidic acis and levofloxin, and more than 50% *Listeria* strains obtained from dairy farms were resistant to the cited antibiotics [178].

In the case of *L. monocytogenes* strains obtained from groundwater in agricultural areas of South Africa, a very high percentage of strains were resistant to tetracycline (90%), doxycycline (85%), cefotaxime (80%), penicillin (80%), chloramphenicol (70%), linezolid (65%), erythromycin (60%) and trimethoprim/sulfamethoxazole (55%). It should be noted that most of the isolates showed multi-drug resistance to at least one antibiotic of three or more antimicrobial groups, and the MAR (multiple antibiotic resistance) indices of all multi-drug resistant isolates were 0.2 [167]. *Listeria* spp. strains isolated from pigs and from a slaughterhouse in Brazil showed a high rate of resistance to ceftazidime (100%), clindamycin (50%), daptomycin (80–100%), and fluoroquinolones, including ciprofloxacin (10–100%), as well as nitrofurantoin (50–57%) and oxacillin (20–50%) [179].

One of the latest studies, analysing drug resistance among 120 *L. monocytogenes* strains isolated in Poland [180] from a total of 6000 samples from pigs, cattle, and poultry, confirmed resistance to cotrimoxazole (45.8%), meropenem (43.3%), erythromycin (40.0%), penicillin (25.8%), and ampicillin (17.5%), with a high level of multi-drug resistance. In Germany [181], among 259 L. monocytogenes strains isolated from food and processing environments and samples from patients, as many as 145 strains revealed multidrug resistance (resistance to ≥3 antibiotics). The strains mainly showed resistance to daptomycin (100%), tigecycline (40%), tetracycline (23%), and to a lesser extent to ciprofloxacin (10%), ceftriaxone (8%), trimethoprim/sulfamethoxazole (6.5%) and gentamicin (4.6%).

**Table 3 antibiotics-11-01079-t003:** Prevalence of specific genes of resistance to selected antibiotics in *Listeria* spp.

	Antibiotic Resistance Genes		Source of Isolation of the Strains and Percentage (%) of Positive Isolates Showing the Presence of Resistance Genes							References
			Dairy Farms	Environment	Water Environment	Raw Fish	Food Products, Dairy, Poultry and Pigs	Poultry Farm and Slaugterhouses	Humans	
*L. monocytogenes*	Beta-lactamases	*penA*	37	-		11.6	-	-	-	Srinivasan et al. [176]; Jamali et al. [182]; Haubert et al. [183]; Iwu & Okoh [167]; Kayode et al. [177]; Oswaldi et al. [184]
		*ampC*	-	0	63	14	-	0	0	
		*bla_TEM_*	-	10	75	-	-	-	-	
		*bla_z_*	-	5	10	-	-	-	-	
	Tetracyclines	*tet(A)*	32	0	85.2	23	-	35.7	0	Srinivasan et al. [176]; Kayode et al. [177]; Iwu & Okoh [167]; Oswaldi et al. [184]; Jamali et al. [182]; Davanzo et al. [185]; Bae et al. [186]; Heidarzadeh et al. [187]; Hanes and Huang [175]; Hailu et al. [178]
		*tet(B)*		19	38	3.3				
		*tet(C)*		17	63	25		7.86	72	
		*tet(M)*	-	4–18	70.37	25.6	52.6	14.3	70	
		*tet(O)*	8					8		
		*tet(S)*					9.1			
	Quinolones	*strA*	34	9		0	-	0	-	Srinivasan et al. [176]; Hailu et al. [178]; Kayode et al. [177];
		*aadA*	51.9					50		
		*aadB*								
	Sulphonamides	*dfrD*	16	11	-	-	-	-	27.3	Kayode et al. [177]; Hanes and Huang [175]; Iwu & Okoh [167];
		*sul1*	16	3.33	38.24	0	0	0	13.6	
		*sul2*	-	13.33	41.18	-	-	-	-	
	Aminoglicosides	*ant6*					18.2			Oswaldi et al. [184]; Iwu & Okoh, [167]; Kayode et al. [177];
		*aadA*	-	12.5	20	-	-	-	-	
		*strA*	-	-	43.33	-	-	-	-	
	Lincosamides	*fosX*		100			100		72–97	
		*vgaD*		100	13		100			
	Macrolides	*ermB*	-	42	-	-	4	14.3	83.3	Haubert et al. [183]; Kayode et al. [177]; Davanzo et al. [185]; Heidarzadeh et al. [187]; Hanes and Huang [175];
	Glicopeptides	*vanA*	-	0	0	4.65	0	-	-	Jamali et al. [182];
	Phenicoles	*floR*	66	4	0	-	-	-	0	Srinivasan et al. [176]; Kayode et al. [177];
		*catI*	-	-	53.3	-	-	-	-	

*ermB*—erythromycin; *fosX*—phosphomycin; *vgaD*—lincosamides, *ant6—*streptomycin; *tet*—tetracyclines, doxycycline, tetracycline, and minocycline; (-)—not detected.

### 3.6. Salmonella spp.

*Salmonella* infections are a serious epidemiological and economic problem all over the world in the context of food safety and public health. *Salmonella* bacteria also are among the most common human foodborne pathogens in the European Union [188]. In 2014 in the EU a total of 88,715 cases of salmonellosis were reported in humans, of which 34.4% involved hospitalization [189]. Some serotypes also become localized in the reproductive tract.

In humans, infected with Salmonella the drug of choice is a third-generation cephalosporin, e.g., ceptriaxone. After confirmation of complicated salmonella or typhoid infection, fluoroquinolones (ciprofloxacin, ofloxacin or fleroxacin), azithromycin or ceftriaxone are used. If treatment is not working as expected or if the isolated strains of Salmonella are resistant to commonly used antibiotics, carbapenems (ertapen or meropen) may be necessary. Patients with complicated infections usually require 7 to 10 days of antibiotics, but typhoid fever usually takes 10–14 days of treatment. In carriers, treatment can last up to four weeks (or longer), and fluoroquinolones are used for this purpose [190]. In animals the using of Trimethoprim-sulfonamide combinations in Salmonella infections shows a positive therapeutic effect. Alternatives for this treatment is using of ampicillin, fluoroquinolones, or third-generation cephalosporins which should be continued daily for up to 6 days [191].

Prevention and control of Salmonella infections in Western Europe and North America is connected to the introduction of treatment of municipal water, pasteurization of dairy products, and exclusion of human feces from food production. Because treatment of the intestinal form of salmonellosis or asymptomatic infected individuas is a controversial issue due to the risk of emergence of carriers of these bacteria or the development of *Salmonella* resistance to the antibiotics used. Therefore, in EU countries, national programs for the control of certain *Salmonella* spp. have been introduced, to control and reduce the level of infection in farm animals, especially poultry and pigs according to the Regulation of the Minister of Agriculture and Rural Development [192]. The primary goal of such programs is to prevent Salmonella infections in farm animals. Since salmonella are facultative intracellular bacteria, the best protection is obtained with live attenuated vaccines which used in pigs, cattle and chickens stimulate a strong cellular immune response and protect the animals from both systemic infection and intestinal colonization. A live attenuated vaccine containing the *S.* Choleraesuis serovar used in pigs appears to be effective in reducing tissue colonization and protecting from disease following experimental infection with virulent Salmonella strains under field conditions [193].

Also, in the case of developing countries, the strategies for enteric fever prevention include improving sanitation, the safety of food and water supplies, identification and treatment of chronic carriers of *Salmonella* ser. Typhi, and the use of typhoid vaccines. The important treatment is also reduction of the proportion of people without access to drinking water sources [190].

The most worrisome phenomenon is multidrug resistance, which limits the choice of antimicrobials in treatment.

Mechanisms of resistance of *Salmonella* to antibiotics can be classified as modifications of the action of an antimicrobial agent or its destruction, or active removal of the antibiotic from the cell (efflux), a mechanism of medical importance. This system is expressed in many clinically important bacteria. The genome of *Salmonella enterica* serovar Typhimurium contains transport proteins AcrA and AcrB (with very high homology to proteins AcrA and AcrB in *Escherichia coli*) and AcrD and Arf [194]. Transport of quinolone antibiotics, tetracycline and chloramphenicol depends on protein AcrB [195].

Acquired resistance in *Salmonella* spp. may be caused by structural or regulatory mutations in chromosomal genes and/or acquisition of new genetic sequences transported onto mobile elements [196]. Both mechanisms of acquisition of genetic variation can cause sudden changes in a population of bacteria and thus affect the evolution of resistance in *Salmonella* spp. Following conjugation, mobile elements of DNA can be maintained as extrachromosomal plasmids or can be completely or partially incorporated into the bacterial chromosome and function as genomic islands [197]. The presence of one resistance mechanism does not guarantee the survival of bacteria, so the simultaneous occurrence of several different resistance mechanisms to different groups of antibiotics is common [198]. In the case of *Salmonella*, resistance to aminoglycosides, β-lactam antibiotics, chloramphenicol, quinolones, tetracyclines, sulphonamides, trimethoprim, and increasingly, colistin is the most commonly observed.

#### 3.6.1. Resistance to Aminoglycosides

Aminoglycosides are included among antibiotics active against gram-negative rods and some strains of *Staphylococcus aureus*, *S. epidermidis*, *Pseudomonas aeruginosa* and *Mycobacterium tuberculosis* [199]. The main mechanism of action involves disturbance of translation by binding of the antibiotic to the bacterial 30S ribosomal subunit and/or binding to ribosomal proteins, resulting in the death of the cell. Resistance of *Salmonella* to aminoglycosides is determined in part by enzymatic inactivation of the antibiotic through chemical modification involving enzymes catalysing reactions such as acetylation, phosphorylation and adenylation [200]. Genes responsible for the production of aminoglycoside-modifying enzymes are most often located on mobile genetic elements such as plasmids and transposons, which allows them to spread between bacteria [201]. The most commonly described gene determining the production of aminoglycoside-modifying enzymes is the gene of resistance to tobramycin, kanamycin and amikacin *aac*(6′)-Ib [202].

#### 3.6.2. Resistance to β-Lactams

Resistance to β-lactam antibiotics can also be determined by several mechanisms. The first, occurring in both gram-positive and gram-negative bacteria, is associated with penicillin-binding proteins (PBPs). The number of PBPs and their affinity for specific drugs differ between genera of bacteria, resulting in differences in the effectiveness of β-lactams against specific microbes. Β-lactam antibiotics form stable bonds with these proteins and block their enzymatic activity, leading to disturbances in the structure of peptidoglycan, followed by lysis of the bacterial cell. Β-lactamases are encoded by genes (*bla*TEM) located in the bacterial chromosome or on mobile genetic elements such as plasmids, transposons or integrons. These mobile genes can be transferred between strains, species or even genera of bacteria. There are two kinds of resistance mechanisms dependent on PBPs. The first involves modification of natural PBPs so that they lose their affinity to β-lactams, but at the same time serve as catalysts in cell wall production. The second type involves acquisition of a foreign, complete gene encoding a PBP which does not react with β-lactam molecules [203]. Two more resistance strategies consist in limiting the ability of a β-lactam antibiotic to penetrate the bacterial cell. This involves reducing the number of porin channels in the outer membrane of bacteria, thereby limiting the ability of β-lactam molecules to enter the cell space. Another mechanism involves actively pumping the drug out of the bacterial cell, which can be intensified by an increase in the expression of efflux pump and porin systems, which are responsible for this process [204]. Another mechanism involves production of β-lactamases, which are specific enzymes catalyzing the hydrolysis of the β-lactam ring in the molecule of the drug [205]. Most gram-negative rods have their own β-lactamases typical for a given species, whose genes are located in chromosomal DNA. A second group of enzymes is extended-spectrum β-lactamases (ESBL), which are mainly acquired, plasmid-encoded β-lactamases. Genes encoding ESBL are often located on conjugative plasmids, including those with a broad host range, so that they are rapidly spread, including between strains of different species. Bacterial strains capable of ESBL synthesis are considered “alert pathogens”. Genetic analysis of bacterial strains and resistance genes in farm animals, food, and humans has shown strong similarities and common genetic traits, providing evidence that ESBL genes, mobile genetic elements and resistant strains are transferred to humans via the food chain [206].

#### 3.6.3. Resistance to Phenicols

Phenicols are broad-spectrum antibiotics whose mechanism of action involves reversible binding to the bacterial 30S ribosomal subunit and selective inhibition of the activity of the enzyme peptidyl transferase, blocking biosynthesis of the protein in the ribosomes [207]. *Salmonella* bacteria have two mechanisms of resistance to chloramphenicol. The first involves enzymatic inactivation of the drug through acetylation by various types of phosphotransferases and chloramphenicol acetyltransferase. The gene of resistance to chloramphenicol is located in a plasmid. Other mechanisms of resistance of *Salmonella* to chloramphenicol are active pumping of molecules of the antibiotic out of the bacterial cell by transporter proteins (efflux pumps); changing the permeability of outer cell membranes; and chromosome mutations which alter the protein of the 50S ribosomal subunit. *S*. Typhi isolates have been shown to possess *cat* genes, which are transferred by a plasmid [208]. The presence of the *cat1* and *cat2* genes has also been detected in other *Salmonella* serovars, such as Derby, Hadar, Enteritidis and Typhimurium [209]. The occurrence of other closely related genes, *cmlA* and *floR*, which encode efflux pumps for chloramphenicol and florfenicol, has also been described in *Salmonella* [210]. The presence of the gene *floR* has been detected on genomic islands and in plasmids in various serovars of *Salmonella*: Agona, Kiambu, Albany, Newport, Typhimurium, and Typhimurium var. Copenhagen [211,212].

#### 3.6.4. Resistance to Tetracyclines

Tetracyclines are broad-spectrum antibiotics which are effective against many gram-positive and gram-negative bacteria, chlamydia, mycoplasmas, and even some protozoa. They act mainly by stopping the binding of tRNA to the A site of the 30S ribosomal subunit and inhibiting protein synthesis [213]. Resistance to tetracycline can be ascribed to a pump removing the antibiotic from the interior of the bacterial cell. The most common *tet* genes of resistance to tetracyclines belong to classes A, B, C, D and G [214] and are located both in plasmids and on the chromosome. Owing to their location in mobile genetic elements, they are easily transferred between strains and are widespread among *Salmonella* bacteria. Most genes are present in multi-drug resistant isolates, which makes them an important marker in identification of potentially important *Salmonella* infections [215].

#### 3.6.5. Resistance to Colistin

Resistance of *Salmonella* to colistin is encoded by *mcr* (mobile colistin resistance) genes, which are located on mobile genetic elements in plasmid DNA and can be easily transferred to other bacteria during cell division or horizontal gene transfer (conjugation or transduction). Five different *mcr* genes have thus far been described: *mcr-1* to *mcr-5* [216,217].

#### 3.6.6. Resistance to Sulphonamides

Sulphonamides are a group of organic chemical compounds which are amides of organosulphonic acids. They are bacteriostatic drugs whose mechanism of action involves competitive inhibition of enzymes taking part in synthesis of tetrahydrofolic acid [213]. Resistance of *Salmonella* to sulphonamides has been ascribed to the presence of an additional plasmid gene, *sul*, which determines expression of the inactive form of dihydropteroate synthase (DHPS) [218]. Another gene coding for ‘normal’ (unmodified) DHPS is present on the chromosome in both resistant and susceptible bacteria. Plasmid-encoded DHPSs are 1000 times less susceptible to sulphonamides than DHPSs encoded by a chromosomal gene. Three genes have thus far been identified: *sul1*, *sul2* and *sul3*. The gene *sul1* is present in many *Salmonella* serovars: Enteritidis, Hadar, Heidelberg, Orion, Rissen, Agona, Albany, Derby, Djugu and Typhimurium [218]. Similarly, resistance to trimethoprim is mediated by genes coding for variants of dihydrofolate reductase (*dhfr* and *dfr*) with reduced affinity for the chemotherapeutic.

#### 3.6.7. Resistance to Quinolones

Among *Salmonella* strains resistant to quinolones and/or fluoroquinolones, the most commonly identified mechanism is the substitution of amino acids in quinolone resistance-determining regions (QRDR) in genes *gyrA* and *parC*. However, plasmid-mediated quinolone resistance genes (PMQR), i.e., the *qnr* genes (*qnrA*, *qnrB*, *qnrC*, *qnrD*, and *qnrS*), are increasingly identified. *Salmonella* can acquire resistance by protecting the target site of the antibiotic, which can be an enzyme or a specific cell structure. For example, the plasmid-encoded quinolone resistance protein (Qnr) confers resistance by acting as a DNA homologue, which competes for binding of DNA gyrase and topoisomerase IV [219], making it more difficult for the quinolone molecule to bind to DNA gyrase and thus protecting the bacteria against the antibiotic. *Salmonella* can also modify the target site of the antibiotic to avoid binding. For example, resistance to rifampicin is based on single-step point mutations, which cause amino acid substitutions in the *rpoB* gene, reducing the drug’s affinity for DNA-dependent RNA polymerase and allowing *rpoB* transcription to continue [9]. The widespread use of fluoroquinolones in the poultry industry is a major problem. Quinolone-resistant bacteria spreading through ingestion of contaminated food have been shown to affect treatment of infections in human [189].

Analysis of resistance to antibiotics in *Salmonella* strains isolated from chicken eggs and from samples collected from laying hens in North and South America, Africa, Europe and Asia in 2013–2019 showed that the most commonly isolated serovars were *S*. Enteritidis and *S.* Typhimurium. The high prevalence of *S.* Enteritidis in eggs and samples from laying hens is likely due to the ability of this serovar to colonize tissues of the reproductive system (ovary and oviduct) [220]. The highest percentages of strains (31.9–40.3%) exhibited resistance to the β-lactam antibiotics ampicillin and amoxicillin/clavulanic acid and to nalidixic acid (a fluoroquinolone). A much smaller percentage of strains were resistant to aminoglycoside antibiotics: gentamicin (12.5%) and streptomycin (18%). Resistance to ciprofloxacin (a fluoroquinolone) was 7.6%. No resistance to antibiotics commonly used in human medicine was detected, i.e., ceftriaxone, cefotaxime, ceftazidime and cefepime, or to imipenem, meropenem and aztreonam, which according to the WHO are high priority critically important. In addition, no strains were found to be resistant to chloramphenicol or trimethoprim/sulfamethoxazole. In the case of trimethoprim/sulfamethoxazole, the low level of resistance is probably due to the fact that sulphonamides are not widely used in laying hens, in contrast to chicken broilers [221,222].

Strains of *S.* Enteritidis and *S.* Typhimurium obtained from broiler chickens in various geographic regions showed a high level of resistance to nalidixic acid (80.3%), ampicillin (64.8%), streptomycin (33%), amoxicillin/clavulanic acid (29.4%) and trimethoprim/sulfamethoxazole (39.3%). Levels of resistance to ciprofloxacin (19%), chloramphenicol (13.6%) and gentamicin (6%) were relatively low, most likely because gentamicin and chloramphenicol are no longer used to treat diseases in poultry [223,224,225,226,227]. Resistance of strains isolated in European countries and the USA has generally been lower, which is linked to the establishment of institutions and implementation of programmes for monitoring antimicrobial agents in poultry production, i.e., the European Food Safety Authority and the European Centre for Disease Prevention and Control, and the National Antibiotic Resistance Monitoring System in the USA [228].

### 3.7. Staphylococcus spp.

*Staphylococcus* spp. is the most common in human and animal bacteria causing the skin and mucosae clinical infections, such as pyoderma, otitis, soft tissue infection and surgical wound in-fections including *Staphylococcus aureus* cased bacteremia (SAB) A factor increasing the importance of bacteria of the genus *Staphylococcus* in the pathology of mammals and birds is their resistance to numerous antimicrobial agents [229]. Staphylococci are sensitive to penicillins, but due to the emergence of strains resistant to natural penicillins, semi-synthetic penicillins such as oxacillin, nafcillin and cloxacillin are used in therapies. Vancomycin is the second important antibiotic used as a basis especially for fighting infections caused by methicillin resistant *Staphylococcus* spp. However, considering the potential risk of resistance occurrence to vancmycin (there are already vancomycin-resistant *Staphylococcus* strains in the world, also in Poland), the so-called alternative to vancomycin, show good activity against resistant staphylococci including: trimethoprim-sulfamethoxazole, ceftaroline, daptomycin, fosfomycin, linezolid, oritavancin/dalbavancin, telavancin or omadacycline [230,231]. Vancomycin therapy has a specific risks especially renal dysfunction, however, vancomycin is still the standard of the treatment of resistant staphylococcal infections and is called also as the last chance antibiotic [229].

In animals, especially livestock, the therapy of infections caused by *Staphylococcus* spp. is corelated with the kind of health problem. For example in North America in dairy cows for the treatment of bovine mastitis are most often used cephapirin, pirlimycin and ceftiofur [232]. In European Union countries the using of antibiotics usage for therapeutic purposes is particularly high in pigs and poultry and less in cattle and sheep. In case of skin infections in livestock, the most commonly used antimicrobial agents in livestock are tetracyclines and penicillins. However, the veterinarian also use macrolides, aminoglycosides and fluoroquinolones. For therapeutic purposes to treat bovine mastitis, pneumonia in calves, metritis in cows and erysipelas in pigs β-lactams, especially penicillins, are most often used [233,234].

Livestock have been identified as main reservoir of multi-drug resistance *Staphylococcus* strains. Bacteria with this resistance profile have been classified as MDR (multi-drug resistance), XDR (extensive drug resistance) and PDR (pandrug resistance). MDR refers to strains that are not susceptible (with resistance or intermediate susceptibility) to at least one antibiotic from at least three groups. XDR refers to bacteria that are non-susceptibility to at least one agent in all but two or fewer antimicrobial categories. PDR indicates total resistance to all antibiotics from all antimicrobial groups. Due to very high genetic and phenotypic variation and to adaptation to the conditions of the environment, staphylococci have become resistant to most currently used antimicrobials. [235].

#### 3.7.1. Resistance to Tetracyclines

In the case of tetracyclines, one of the mechanisms of resistance of staphylococci involves a decrease in their intracellular concentration due to a specific mechanism of removal (efflux), associated with the presence of the genes *tet*K and *tet*L, encoding membrane transporters [236]. Inducible resistance to tetracyclines is encoded by small plasmids, whereas constitutive resistance is encoded by chromosomal determinants *tet*(M) and *tet*(O) and is not associated with an efflux pump but only with active protection of the ribosome against binding to tetracycline [237].

Active removal of the antibiotic from the cell is also a mechanism of resistance to fluoroquinolones and the trimethoprim/sulfamethoxazole combination.

#### 3.7.2. Resistance to β-Lactams

The main mechanisms of resistance to β-lactam antibiotics are the ability to produce β-lactamase, resulting in resistance to natural penicillins, amino- and ureidopenicillins, and production of penicillin-binding protein PBP2a, also called PBP2′, with low affinity for β-lactam antibiotics, resulting in resistance to all β-lactam antibiotics currently used in treatment [238]. The first epidemics induced by hospital-acquired methicillin-resistant *Staphylococcus aureus* strains appeared in the late 1970s and early 1980s. These bacteria, produce a modified penicillin-binding protein (PBP2a) to prevent β-lactam antibiotics from binding to the target site of action, causing compounds whose structure contains a β-lactam ring to become ineffective. Resistance to β-lactam antibiotics is determined by the gene *mec*A or *mec*C, located in the chromosome and constituting part of the region known as staphylococcal cassette chromosome mec (SCCmec). Both *mec*A and *mec*C are responsible for synthesis of a modified protein that does not allow binding by penicillins, cephalosporins (except for the latest generation, i.e., ceftaroline), carbapenems or monobactams. A new homologue of *mec*A (*mec*ALGA251), i.e., *mec*C, has been detected in bacteria isolated from humans and farm animals in many European countries, as well as in pets and wild animals. Cattle populations have been recognized as the main reservoir of *mec*C strains [238]. Interestingly, the MECC gene has rarely been reported in bird species. It is estimated that in in humans, about 80–90% of isolates associated with hospital-acquired infections are methicillin-resistant coagulase-negative staphylococci (MRCNS). It has been postulated that *S. epidermidis* may function as a reservoir of genes for the more pathogenic species *S. aureus*. Moreover, results reported by other authors indicate that MRCNS are isolated from food more often than methicillin-resistant *S. aureus* [239].

#### 3.7.3. Resistance to Methicillin Macrolides Lincosamides and Streptogramin B

It should be noted that MLSB resistance in *Staphylococcus* spp. refers to resistance to macrolides, lincosamides and streptogramin B. The mechanism of action of this group of antibiotics involves inhibiting synthesis of proteins at the level of the 23S rRNA subunit, where they bind to adenine at position 2058 or 2059 in domain V. The *erm* genes encode ribosomal methylase modifying adenine at the target site of the antibiotics in the 23S rRNA subunit, which blocks them from binding to the bacterial cell [240]. Many types of genes of this group have been described in staphylococci: *erm*(A), *erm*(B), *erm*(C), *erm*(F), *erm*(G), *erm*(Q), *erm*(T), *erm*(Y), *erm*(33), *erm*(43) and *erm*(48) [241]. There are two types of MLSB resistance: constitutive and inducible. In the case of constitutive resistance, the bacteria have active mRNA which enables methylase synthesis without an inductor. In induced MLSB resistance, inactive mRNA is synthesized and is activated in the presence of an inductor. Among MRSA strains derived from farm animals, the presence of more than one *erm* gene has been described, most commonly *erm*(A) with *erm*(C) or *erm*(A) with *erm*(B) [242,243].

Methicillin-resistant strains also usually have other resistance genes (e.g., determining resistance to sulphonamides, gentamicin, kanamycin, streptomycin macrolides, fluoroquinolones or tetracyclines), so that they can be classified as MDR [244]. A widespread mechanism of resistance to aminoglycosides among *S. aureus* is the synthesis of transferases known as aminoglycoside-modifying enzymes (AME): acetyltransferase [(AAC (6′)/APH(2″)] encoded by *aac*(6′)/*aph*(2″), phosphotransferase [APH(3′)-III, ANT (4′)-I ] encoded by *aph* (3′)-IIIa, *ant*(4′)-Ia; and nucleotidyltransferase [ANT (9)-I] encoded by *ant*(6)-I [85,245]. The enzymes modify the antibiotic molecule and thereby deprive it of its antimicrobial activity. Reduced susceptibility or resistance can also be due to disturbed transport of aminoglycosides to the interior of the bacterial cell. Clinically the most common AMEs in staphylococci are ANT (4′)-I, AAC (6′)/APH(2″) and APH (3′)-III, which modify aminoglycosides of therapeutic importance, including tobramycin, gentamicin and kanamycin [246].

#### 3.7.4. Resistance to Fluoroquinolones

An important mechanism of resistance to fluoroquinolones in staphylococci is a mutation of topoisomerase II (gyrase) and topoisomerase IV—bacterial enzymes involved in replication of the target DNA for fluoroquinolones. Each of them contains two subunits, *gyr A* and *gyr B* and *par C* and *par E*. Disturbance of synthesis of one enzyme or both at the same time leads to inhibition of replication of the DNA of the cell and its death. This region of mutation in the gyrase and topoisomerase genes is called the quinolone resistance-determining region (QRDS). Mutation in the QRDS results in the replacement of one amino acid with another at a specific active site of the enzyme [247].

Another mechanism is the acquisition of genes of drug resistance by importing them from outside. This type of resistance is called plasmid resistance [248]. Due to the autonomy of plasmids, they can be transferred to other bacterial cells without the involvement of a chromosome. The co-occurrence of chromosomal and plasmid resistance means that the drug resistance of bacteria to specific antimicrobials develops as a result of activation of various mechanisms, depending on the drug. An important element of resistance of *S. aureus* to fluoroquinolones is the presence of an efflux pump mechanism. In staphylococci, overexpression of proteins forming three efflux pump groups, Nor A, Nor B and Nor C, causes a 4–8-fold increase in the MICs for fluoroquinolones. Transporter Nor A is responsible for resistance to hydrophilic fluoroquinolones, such as norfloxacin and ciprofloxacin, and Nor B and Nor C for resistance to fluoroquinolones, both hydrophilic and hydrophobic, including moxifloxacin [249]. Resistance to moxifloxacin is only half as high as resistance to ciprofloxacin, so it is particularly recommended in treatment of *Staphylococcus* infections. Resistance to fluoroquinolones among hospital-acquired strains of *S. aureus* is usually found together with methicillin resistance.

The gene *cfr* encodes methylase, with activity against adenine at position 2503 in domain V of 23S rRNA [250]. This is the target site of phenicols, lincosamides, oxazolidinones, pleuromutilin and streptogramin. Methylation of the target site causes cross-resistance to all of these antimicrobials. The *cfr* gene was first described in 2000 on plasmid pSCFS1 detected in a *S. scuiri* isolate from cattle [251]. It is located on plasmids which usually additionally contain genes of the *erm* group, also determining multi-drug resistance [252].

Reports by the World Health Organization (WHO) indicate that the latest data on the occurrence of various species of staphylococci in livestock and their resistance to antibiotics differ depending on geographic location, animal species, and housing system, and are also associated with the level of economic development of a given part of the world. In comparison with northern Europe, there were generally more cases of resistance in southern and south-eastern parts of the continent. In 2013–2016 the percentage share of MRSA (methicillin-resistant *S. aureus*) significantly decreased in EU countries. Despite this favourable trend, MRSA remains a priority issue for public health in Europe, as 10 of 30 countries have reported a MRSA percentage over 25% [253].

A study published in 2020 on the occurrence of *Staphylococcus* spp. In pigs in Greece showed their presence in 48.61% of samples. The dominant species was *S. aureus*, while the dominant coagulase-negative species were *S. epidermidis* and *S. saprophyticus*. These strains showed high resistance to tetracycline (97.1%) and clindamycin (80.0%), but a much lower percentage were resistant to fusidic acid (14.3%). No *S. aureus* strains resistant to methicillin (MRSA) or vancomycin were identified [254]. In a study on *Staphylococcus* in pigs in India, the most commonly isolated species were *S. sciuri*, *S. aureus*, *S. lentus* and *S. hyicus*, with most isolates showing multi-drug resistance. The highest percentage of strains showed resistance to ampicillin and penicillin. There were also strains resistant to vancomycin, and as many as 66.67% of *S. aureus* isolates were resistant to methicillin.

Transmission of drug-resistant strains from animals to humans usually takes place through consumption of contaminated animal products (meat or eggs). Examination of samples taken from chicken broilers during slaughter in Morocco showed that more than half of isolated *S. aureus* strains (54%) were resistant to penicillin, 29.4% to tetracycline, 23.5% to erythromycin, and 17% to ciprofloxacin [255]. Most *Staphylococcus* spp. Isolates obtained from hens from commercial flocks in Pakistan were resistant to the tetracycline derivative tigecycline (74.8%), while among staphylococci isolated from backyard hens the highest level of resistance was to clindamycin (72.1%). The presence of the *mec*A gene was detected only in strains from backyard hens, whereas the genes *erm*C and *tet*(K) or *tet*(M) were identified in *Staphylococcus* strains from both groups. Multi-drug resistance was observed in 88% of strains. These results also reflect the influence of the environment or habitat of birds in different rearing systems on the intestinal microbiota [256].

## 4. Conclusions

Acquisition of resistance to antibiotics by bacteria is one of the most important problems of modern medicine. A particularly disturbing phenomenon is the prevalence of very high percentage, in many cases even 100% of multi-drug resistant foodborne pathogens in developing countries, mainly in Africa and Asia. It has also been reported that, in many bacterial species, the acquisition of drug resistance is mediated by the interspecies transfer of resistance genes through the resistance mechanism. Such a transfer, among others by plasmid transfer by conjugation confirmed in numerous studies in *Enterococcus* spp., *Styaphylococcus* spp., *Streptococcus* spp., *Listeria* spp. indicates a significant global risk of continuous increase especially multi-drug resistance among bacteria.

Therefore, there is a need to monitor the resistance of these bacteria which will allow you to control the extent of the spread of drug resistance among bacteria on the world. The occurrence of a variety of bacterial species in farm animals and their resistance to antibiotics differ depending on geographic location, the animal species, and housing system, and is also associated with the level of economic development of a given part of the world

Such a quick and easy spread of multidrug resistance among bacteria is a global threat to humans and animals and imposes an obligation not only in the field of comprehensive diagnosis of drug resistance, but also in the implementation of methods of bacterial control alternative to antibiotics. Therefore, there is a necessity to develop of novel strategies in control of bacterial infectious is high demand. In response, several new therapies have been tested with using of bacteriophages, antimicrobial peptides, and combinations of two or more antibiotics in therapy.

However, due to the fact that most infections occur through the alimentary tract, it is necessary to take preventive measures such as improving sanitation, ensuring the safety of food and water supplies, rapid identification and treatment, and the development of new generation vaccines to reduce the susceptibility to infection.

## Figures and Tables

**Table 2 antibiotics-11-01079-t002:** The examples of the prevalence of antibiotic resistance gens in *E. coli* strains isolated from the environment and different animal species.

	Antibiotic Resistance Genes		Source of Isolation of the Strains and Percentage (%) of Positive Isolates Showing the Presence of Resistance Genes							
			Humans	Ruminants	Pigs	Poultry, Wild Birds	Companion Animals (Cats, Dogs, Horses, Pet Birds)	Food	Environment (Surface Soil, Sewage, Drinking and Pond Water)	References
*Escherichia coli*	Beta-lactamases	*bla* _CTX-M_	96.6	46.5	-		34.65–48.9	10.1	-	Bahramian et al. [111]; Maynard et al. [148]; Sheikh et al. [130]; Tian et al. [149]; Cormier et al. [150]; Gundran et al. [151]; Wang et al. [152]; Chen et al. [153]; Ilyas et al. [154]; Ejikeugwu et al. [112]; Ombarak et al. [155]; Mahmoud et al. [156]; Nowaczek et al. [135]
		*bla* _TEM_	58.6	56.5–97.1	86	24–57.97	17–95.28	0.8–18	-	
		*bla* _SHV_	-	16.0	21	27.5	16.55	1.2–2.0	-	
		*bla* _OXA_	-	-	5	-	7.09–14.02	-	15.5	
		*bla* _CMY_	72.4	88.4	-	-	35.9–9.45	2.6–14.7	-	
		*bla* _DHA_	20.7	-	-				-	
		*bla* _ACC_	37.9	-	-	-	-	-	-	
		*bla* _IMP_	72	16.7	-	10.2	-	-	-	
		*bla* _VIM_	28	-	-	23.7	-	-	-	
		*bla* _NDM_	4–51.7	-	-	31.8–19.8	2.36	-	-	
		*bla* _KPC_	22.4	-	-	-	-	-	4.4	
	Tetracyclines	*tet(A)*	32.2	76.7–51.1	25–57.7	12.5–52.4	38.8–18	26.8–23.8	-	Jahantigh et al. [131]; Maynard et al. [148]; Belaynehe et al. [157]; Ombarak et al. [155]; Schwaiger et al. [158]; Gholami-Ahangaran et al. [159]; Ahmed et al. [137]; Nowaczek et al. [135]
		*tet(B)*	55.9	23.3–44.6	80–38.7	41.3	61.1–71	23.2–4.05	-	
		*tet(C)*	7.2	5.4	25–5.1	1.7	-	4.3	-	
		*tet(D)*	1.3	-	2.9	0.8	-	0.4	-	
		*tet(E)*		-	-	-	-	-	-	
		*tet(G)*		6.5	-	-	-	-	-	
		*tet(M)*	6.6		2.9	-	-	-	-	
	Quinolones	*qnrA*	-	-	2.0	-	17.32	-	0.4	Chen et al. [117,153]
		*qnrB*	0.3	-	-	1.3	93.70	-	1.1	
		*qnrS*	2.6	-	8.6	1.3	8.66	-	4.2	
		*qepA*	3.6	-	4.5	1.3	-	-	2.6	
		*oqxAB*	5.2	-	51.0	19.8			20.2	
		*aac*(3)-IIa	70.7	80	-	-	1	-	-	Cirit et al. [125]; Maynard et al. [148]; Sheikh et al. [130]; Yu et al. [121]; Ombarak et al. [155]; Abo-Amer et al. [160]; Schwaiger et al. [158]; Belaynehe et al. [161]; Usui et al. [162]; Sigirci et al. [163]; Nowaczek et al. [135]
		*aac*(3)-IV		13.3	75	20	-		0.7	
		*aac*(6)-Ib	76.9	-	-	-	-	-	-	
		*rmtB*	-	5.3	2.6	-	0.9	-	-	
		*strA/B*	61.2/63.8	76.7	52.6/54.7	-	2/3	18.4	28	
		*aadA*	59.2	71.7	-	20	1	4.95	10.4	
		*aadB*	3.1	-	-	-	1	-	0.5	
		*aphA1*	7.6	-	79	18.7	2	4.05	6.6	
		*aphA2*	-	-	19	-	1	-	1.7	
		*sul*II	61.8	63.3	17.4–40.1	31.1–70.6	25.8	9.4–12.3	-	
		*sul*III	3.3	-	7.6–2.2	-	-	0.9–1.35	-	
		*dhfr*I	-	76.3	20	-	40.3	-	-	
		*dhfr*V	-	15.8	47	-	-	-	-	
		*dhfr*XIII	-	-	30	-	-	-	-	
		*dhfr*IX	-	-	-	-	0.3	-	-	
	Polymyxins	*mcr-1*	-	11.5	4.45–20.6	25.0	2.36	14.9	-	Khine et.al. [144]; Liu et al. [142]; Wang et al. [152]; Chen et al. [153]
		*mcr-2*	-	-	20.8	-	-	-	-	
		*mcr-3*	-	-	0.43	-	-	-	-	
	Phenicoles	*catI*	79	85.1–47.4	-	61.7	73.5	-	-	Maynard et al. [148]; Belaynehe et al. [157]; Abo-Amer et al. [160]; Ibrahim et al. [164]; Derakhshandeh et al. [165]; Ahmed et al. [137]
		*floR*	11.4	1.5–50	-	-	9.7	-	-	
		*cmlA*	-	10.4–18.4	-	72	4.8	-	-	

## Data Availability

Data is contained within this article and Appendix A.

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
