# Peer review of "Antibiotic Resistance in Bacteria—A Review"

_antibiotics, 2022, doi:10.3390/antibiotics11081079_

Round 1
Reviewer 1 Report
The review describes the mechanisms of resistance in most frequent bacteria called as “foodborne pathoges” isolated from people and animals. It is a very deep study, in much parts redudant. I suggest to combine and shorten some paragraphs, such as when the authors described the various types of resistance, and then move to the bacteria repeating the resistance mechanisms. On the other hand, the review lacks of explicative figures. In addition, no mention is for bloodstreem infections (see the article doi: 10.3390/antibiotics9120851). Finally, in the abstract the authors reported that there are also own data in the review. Where? Please correct some errors throughout the text, such as in Table 2 change gens with genes.
Author Response
Dear Reviewer
Thank You very much for Your revision of the manuscript. We have made a proper corrections according to suggestions, however we also decided to use some additionaly corrections according to all Reviewers. We have insert also a supplementary file with additional information about the prevalence of phenotypic resistance and resistance genes including the first detection in presented bacteria in different part of the world. However because of the huge size of manuscript some of suggestions couldn't be implemented in teh text.We hope that in this form the manuscript will be more acceptable.
The spelling mistakes we have also correct .
Thank You very much for Your help and work. We really appreciate Your work and assistance.
We also attach the cited references with our own study which are cited in the text of manuscript.
Chapter: Camplobacter spp.- Wieczorek et al. 2015, Wieczorek et al. 2018
Chapter: Enterococcus spp. Stępień-Pyśniak et al. 2018, Stepień-Pyśniak et al.2021a, 2021 b
Chapter: E coli- Nowaczek et al. 2021
Supl. File: Stępień-Pyśniak et al. 2021
Marek et al. 2018
Reviewer 2 Report
I read the review "Antibiotic resistance in bacteria" and found it a well-articulated and comprehensive review of the topic. The authors provided detailed information on the mechanism of acquired drug resistance in bacteria. Further, mechanisms of transfer of resistance were given with examples of various species of bacteria.
I believe that the reader will benefit if the authors will include more details in the introduction (or a separate small section) on the potential impact of antibiotic resistance in bacteria in relation to humans, livestock, and environmental microbial populations. In addition, there should be some information on the available methods to overcome antibiotic resistance.
There are some spelling mistakes that need to be corrected eg in the title 'rewiev'.
I think it is a good review that can be published with minor corrections.
Author Response
Dear Reviewer
Thank You very much for Your revision of the manuscript. We have made a proper corrections according to suggestions. Because the sugestions were very general, so we decided to use some additionaly corrections. Some of the corrections has been made according to all Reviewers. We have insert also a supplementary file with additional information about the prevalence of phenotypic resistance and resistance genes in presented bacteria in different part of the world. We hope that in this form the manuscript will be more acceptable.
The spelling mistakes we have been also corrected, however sometimes is difficult to find all because of the size of the manuscript.
Thank You very much for Your help and work. We really appreciate Your work and assistance.
Kind regards
The Authors
Author Response
Dear Reviewer
Thank You very much for Your detailed revision of the mauscript. We have made a proper corrections according to suggestions, however we also decided to use some additionaly corrections according to all Reviewers. We have insert also a supplementary file with additional information about the prevalence of phenotypic resistance and resistance genes in presented bacteria in different part of the world. We hope that in this form the manuscript will be more acceptable.
Please find below theanswers to the detailed comments.
Major comments:
1- The main issue related to this manuscript is the lack of sufficient novelty compared with several otherliterature reviews.
2- The whole structure of the manuscript is confusing. For example, why authors put the subtitle “2.1.Horizontal gene transfer” when there is only this paragraph? The different chapters were not organized in the same way; the chapters dedicated to Enterococcus and E. coli were divided into several subtitles according to the antimicrobial groups, while the other chapters were not divided.
A: ½-This part of the manuscript has benn corrected . We also changed the structure of the „introduction” chapter. Also we have made a proper corrections in chapter and now all of theam are very similar. Thank You very much for Your suggestions. We have also added some newest information about the prevalence of resistance in bacteria followed with cited references (2021/22 year). We hope that in this for the manusript has more novelty information.
3- The different chapters should be written according to the same plan. For example, in some sections,the authors presented the prevalences of antibiotic resistance and then described the AMR mechanisms,while in other sections, it is the opposite.
A: We have made a proper corrections according to the suggestions. Now the chapter have a very similar structures including subchapters formula. Thank You very muc for Your comments.
4- The authors should give a brief presentation of the pathogen and the induced diseases for all bacterial agents studied in this manuscript. Authors should give information about the drugs of choice in the treatment of each pathogen.
A: Thank You very much, this information has been added in the text of the manuscript according to the suggestions.
5- The writing is wordy and confusing in many places, the authors repeat the definition/the description of the same mechanism in many paragraphs. The manuscript needs thorough proofreading for sentenceformation and the use of repetitive words.
6- I suggest including an updated figure representing the different AMR mechanisms, to avoid repetition in paragraphs.
7- The section relative to Listeria is well written and structured, however it should be more developed.
A: 5-7-Thank You very much for Your comments. Wa have made a proper corrections and we have completed this section with additional elements concerning resistance to particular groups of antibiotics.
8- Despite the authors cited 217 references, several recent publications have been omitted in the
manuscript. Moreover, there are phrases and concepts without references. In a review, each concept andeach statement reported should have its own reference.
9- Authors missed to talk about new reported AMR genes in Campylobacter (see the publications of Tang et al., 2020, Yao et al., 2020…), and missed also important genes in Salmonella section (blaCTX-M, blaSHV,blaDHA ...).
A: Thank You very much for Your suggestions, we have made a proper correction and we have added the lacking information contained in cited references.
10-The reported data in the manuscript did not cover all the regions worldwide. There is a lack of data from Africa and Asia, where the problem of antibioresistance is widely spread. I suggest including figures showing recent data on the prevalence of antibioresistance in different geographical regions and different strains.
11-The authors can include a table containing the characteristics of publications reporting AMR prevalence and AMR genes, as supplementary file.
A: points 10/11- These information have been added according to the sugestions. We ahve also created a supplementary file with table according to the Reviewer’s suggestion. And this table we also completed with the information of all different parts of the world.
12- The authors should indicate the name of plasmids (e.g. line 672, line 869, and others).
13-The abstract does not adequately give a good “snap shot “of the end point of the whole review.
A: Thank You very muvh for Your suggestions, we have made a proper changes in „abstract” according to the main topic and now is more connected to the whole manuscript.
14-The conclusion is a bit superficial.
A: The conclusion was rewritten and we have added a new more detailed information about the resume resulting from this review.
Minor comments:
1- Lines 10-12: Change the sentence " Widespread resistance to antibiotics among bacteria is the cause of hundreds 10 of thousands of deaths every year. In response to the significant increase..." in the abstract, which is repeated in the introduction (lines 29-31).
A: This sentence has benn corrected according to the suggestions.
2- Italicize the bacterial and genes names throughout the manuscript.
3- Please, revise the nomenclature of some genes (e.g. tet(O), not tetO, blaOXA-61 not blaOXA61…).
A: This corrections has been dome according to the instructions. Thank Yopu very much for Your help.
4- Change the word "chemotherapeutics" by "antimicrobials", throughout the manuscript.
A: This correction has been made according to the suggestions.
5- Change the word "people" by "human "throughout the manuscript.
A: It was corrected according to the suggestion.
6- Line 101: Change "circular pieces of DNA" by "small circular molecule of DNA".
A: It was corrected according to the suggestion.
7- Table 2: Rearrange the table, it is not clear. I suggest to present the % of AMR genes according to each reference separately.
8- Table 2: Change "% percentage" by "percentage (%)".
A: the table 2 has been corrected also according to the other Reviewer’s.
9- Some references are too old, and other reference are not adequate for example: you should change reference 1 by the original publication. The same for the reference 199, and others throughout the manuscript.
A: Thank You very much for Your comment, we have change some references, however some of tchem are very important because contain for example the first detection of resistance or first detection of proper genes prevalence in bacteria.
10- I suggest combining the paragraphs: "3.3.1.-3.3.2.-3.3.3." into one paragraph with the title βlactams resistance.
A: It was corrected according to the instructions.
11- Line 722: Revise the sentence: "multi-drug resistance to three or more antibiotics", which is
wrong. The sentence should be changed by "multi-drug resistance to at least one antibiotic of three or more antimicrobial groups". The same comment for the sentences line 794 and 1084-1085.
A: This sentence has been corrected as suggested, while fragments of the sentences in lines 794 and 1084-85 have been removed elsewhere in order not to repeat the content itself.
12- Table 3: Rearrange the format. Since both Table 2 and 3 are presenting the AMR genes in
strains from different origins, so keep the same format for both of them. For CLIR, you should indicate the AMR gene, not the phenotypic result.
A: The table 3 has been completly rewritten and rebuilt, and now looks similar like two earlier tables. Thank You very much for Your help.
13- Lines 961-963: Adjust the definition of XDR, refer to the publication of Rafailidis and Kofteridis (2022).
A: This sentence was corrected according to the suggestion and references.
14- Line 964: Change "chemical groups" by "antimicrobials groups"
A: This was corrected. Thank You very much for Your suggestion.
Once gain
Thank You very much for Your all comments and suggestions, we really appreciate Your work and help.
Kind regards
The Authors
The Authors
Round 2
Reviewer 1 Report
The manuscript is highly improved and is ready for publication.
Reviewer 3 Report
All revisions have been properly performed.